# Intake of *Lactobacillus delbrueckii* (pExu:*hsp65*) Prevents the Inflammation and the Disorganization of the Intestinal Mucosa in a Mouse Model of Mucositis

**DOI:** 10.3390/microorganisms9010107

**Published:** 2021-01-05

**Authors:** Fernanda Alvarenga Lima Barroso, Luís Cláudio Lima de Jesus, Camila Prosperi de Castro, Viviane Lima Batista, Ênio Ferreira, Renata Salgado Fernandes, André Luís Branco de Barros, Sophie Yvette Leclerq, Vasco Azevedo, Pamela Mancha-Agresti, Mariana Martins Drumond

**Affiliations:** 1Laboratório de Genética Celular e Molecular (LGCM), Departamento de—Genética, Ecologia e Evolução, Instituto de Ciências Biológicas, Universidade Federal de Minas Gerais (UFMG), Belo Horizonte 31270-901, Brazil; fernanda_alima@hotmail.com (F.A.L.B.); lc.luiis@yahoo.com.br (L.C.L.d.J.); camilaprosperic@gmail.com (C.P.d.C.); vivianelimabio@gmail.com (V.L.B.); vasco@icb.ufmg.br (V.A.); 2Departamento de Patologia Geral, Instituto de Ciências Biológicas, Universidade Federal de Minas Gerais, Belo Horizonte 31270-901, Brazil; enioferreira@icb.ufmg.br; 3Departamento de Análises Clínicas e Toxicológicas, Faculdade de Farmácia, Campus da UFMG, Universidade Federal de Minas Gerais, Cidade Universitária, Belo Horizonte 31270-901, Brazil; renatasalgadof@yahoo.com.br (R.S.F.); brancodebarros@yahoo.com.br (A.L.B.d.B.); 4Laboratório de Inovação Biotecnológica, Fundação Ezequiel Dias (FUNED), Belo Horizonte 30510-010, Brazil; sodris2003@gmail.com; 5Faculdade de Minas-Faminas-BH, Medicina, Belo Horizonte 31744-007, Brazil; 6Centro Federal de Educação Tecnológica de Minas Gerais (CEFET/MG), Departamento de Ciências Biológicas, Belo Horizonte 31421-169, Brazil

**Keywords:** recombinant probiotics, DNA delivery, intestinal mucositis, inflammation, technetium-99m, bacterial translocation, gene expression

## Abstract

5-Fluorouracil (5-FU) is an antineoplastic drug that causes, as a side effect, intestinal mucositis, acute inflammation in the small bowel. The Heat Shock Protein (Hsp) are highly expressed in inflammatory conditions, developing an important role in immune modulation. Thus, they are potential candidates for the treatment of inflammatory diseases. In the mucositis mouse model, the present study aimed to evaluate the beneficial effect of oral administration of milk fermented by *Lactobacillus delbrueckii* CIDCA 133 (pExu:*hsp65*), a recombinant strain. This approach showed increased levels of sIgA in the intestinal fluid, reducing inflammatory infiltrate and intestinal permeability. Additionally, the histological score was improved. Protection was associated with a reduction in the gene expression of pro-inflammatory cytokines such as Tnf, Il6, Il12, and Il1b, and an increase in Il10, Muc2, and claudin 1 (Cldn1) and 2 (Cldn2) gene expression in ileum tissue. These findings are corroborated with the increased number of goblet cells, the electronic microscopy images, and the reduction of intestinal permeability. The administration of milk fermented by this recombinant probiotic strain was also able to reverse the high levels of gene expression of Tlrs caused by the 5-FU. Thus, the rCIDCA 133:Hsp65 strain was revealed to be a promising preventive strategy for small bowel inflammation.

## 1. Introduction

Radio- and chemotherapy and the combination of both are widely used for cancer treatment. 5-Fluorouracil (5-FU) is one of the main chemotherapeutic drugs used to treat several types of cancer. This drug is responsible for several adverse effects, such as mucositis, an inflammatory process that affects the entire digestive tract, causing abdominal pain, nausea, and diarrhea, which is the main limiting factor for the continuity and, consequently, efficacy of cancer treatment [1,2,3].

Many strategies have been studied, such as the administration of amino acids [4,5], vitamins [6], antioxidants [7], fatty acids [8,9], and, last but not least, the administration of probiotics, mainly lactic acid bacteria (LAB), such as lactobacilli [10,11], to alleviate the mucositis symptoms. The protective effects of probiotics on the intestinal barrier have been related to their influence on innate and adaptive immunity. Their ability to regulate Toll-like receptors (Tlrs) [12] and customize the composition and activity of the gut microbiota leads them to show many immune and non-immune protective mechanisms [13].

The treatment of intestinal diseases, such as colitis and mucositis, with wild-type probiotics has been reported to show promising results [14,15,16,17]. The probiotics display promising results, as demonstrated in different animal models; however, positive results are more limited when treating human inflammatory diseases. For that reason, more studies should be performed.

Thereby, based on these experiences and along with increasing the beneficial characteristics of probiotics, studies on many strains that express or encode different proteins with anti-inflammatory activities as promising candidates for the treatment of other pathological conditions, especially inflammatory bowel disorders, have been performed [15,18,19,20,21,22]. In this context, studies have been developed using the microbial 65 kDa heat shock protein (Hsp65) of *Mycobacterium leprae* (homolog to mammalian Hsp60) in many different animal disease models, to evaluate either prevention or treatment. For instance, to evaluate the effect of this protein in tuberculosis disease, mice [23] and calves [24] were used. Mouse models were used to study colitis [25], encephalomyelitis [26], lupus [27], and atherosclerosis [28], among others, with excellent outcomes, thus demonstrating the relevance of the Hsp65 protein as a good candidate for treatment and therapeutic uses.

Hsp proteins constitute about 5% of all intracellular proteins of prokaryotic and eukaryotic organisms, with high structural homology between bacterial and mammalian Hsps [29]. They participate in protein folding, degradation of misfolded protein, acting as intracellular chaperones, avoiding undesirable protein aggregation during folding and subunit assemblage [30,31], and provide their clearance and recycling [32,33]. The Hsps are ubiquitous antigens expressed as housekeeping proteins [34] and are up-regulated in inflamed tissues, responsible for responding to stressor agents, such as toxins, oxidative injury, inflammatory processes, and infections, leading the reestablishment of homeostasis [35,36,37].

Hsps, also known as stress proteins, are considered to be conserved proteins during evolution, present in all living organisms. These proteins play an essential role in molecular chaperones, interacting either with proteins tagged for degradation or with foreign polypeptide aggregation [38]. Hsps are up-regulated when cells face stressful situations [39,40,41], such as an infectious disease, presenting immunoregulatory activities [42], and also participate in cell survival signaling pathways [43]. Additionally, these molecules are considered natural adjuvants since they stimulate internalization by scavenger receptors and even the presentation of antigens through Major histocompatibility complex (MHC) molecules, inducing the production of chemokines, pro-inflammatory cytokines, as well as the production of nitric oxide by macrophages and dendritic cells [42,44,45].

The precise role of Hsps in Inflammatory Bowel Diseases (IBDs) is still not clear. Miao and colleagues [46] correlated the severity of the disease with elevated levels of Hsp70 in human patients with ulcerative colitis. In patients with IBD, autoantibodies were detected to Hsp60 and Hsp70, linking the disease pathogenesis to the cross-reactivity between Hsps from eukaryotic and prokaryotic organisms [47].

Recombinant *Lactococcus lactis* NCDO2118, producing Hsp65 protein, was developed by our research group [48]. This recombinant strain showed promising results when it was tested in colitis [25] and experimental autoimmune encephalomyelitis (EAE) [26] mouse models. Thus, the immunomodulatory and anti-inflammatory action of the Hsps molecules was shown.

Regarding the mechanisms associated with the beneficial effects of Hsp65, Gomes-Santos et al. [25] used the engineered *L. lactis* mentioned above, associated with the protection of the gut with increased Il10 levels in the colon and an expansion, in spleen and mesenteric lymph nodes, of CD4^+^Foxp3^+^ and CD4^+^LAP^+^ regulatory T cells, possibly in a dependent effect on Tlr2 and Il10. They also showed a reduction in pro-inflammatory cytokines and maybe the same role could be observed in mucositis since both are injuries of the intestinal tract.

Based on these promising results for heterologous protein production by *L. lactis*, our research group developed a broad-range plasmid called pExu [49] to be used as a DNA vaccine vector. In this approach, the host cells are in charge of protein production, with LAB as the delivery vehicle.

Thus, based on the beneficial action of Hsp65 protein in different diseases, and considering the protective effect of milk fermented by the *L. delbrueckii* CIDCA 133 strain in an intestinal mucositis mouse model previously reported by De Jesus et al. [17], the present study aims to investigate the therapeutic/protective and immunomodulatory effect of recombinant *L. delbrueckii* CIDCA 133 (pExu:*hsp65*) on the harmful effects of 5-FU in the intestinal epithelium.

## 2. Materials and Methods

### 2.1. Bacterial Strains and Plasmids

*Escherichia coli* (*E. coli*) Top10 (Invitrogen) were grown aerobically in Luria–Bertani medium (LB) (Acumedia Lansing, MI, USA) at 37 °C with vigorous shaking. *Lactobacillus delbrueckii* subsp. *Lactis* CIDCA 133, belonging to the culture collection of the Centro de Investigación y Desarrollo en Criotecnología de Alimentos (CIDCA, Facultad de Ciencias Exactas, Universidad Nacional de La Plata, Argentina), were grown in de Man, Rogosa, and Sharpe (MRS) broth (Kasvi, São José dos Pinhais, Brazil) at 37 °C for 16 h in microaerobiosis. When necessary, this medium was supplemented with 2.5 µg/mL of erythromycin (Sigma-Aldrich, St Louis, MO, USA) for the growth of recombinant strains, *L. delbrueckii* CIDCA 133 (pExu:empty), and *L. delbrueckii* CIDCA 133 (pExu:*hsp65*).

### 2.2. DNA Vaccine Construction: Recombinant L. delbrueckii CIDCA 133 (pExu:hsp65)

To construct the DNA vaccine, the functional pVax:hsp65 plasmid [50] was digested with 10 units of BamHI (Biolabs, England) and 20 units of NotI (Invitrogen, Carlsbad, CA, USA) restriction enzymes. A 3.3 Kb fragment corresponding to the *Mycobaterium leprae* hsp65 gene was obtained with cytomegalovirus (CMV) intron A. The same restriction enzymes were used to digest the empty pExu plasmid [49]. The insert and the digested vector were purified using a commercial kit (IllustraTM GFXTM PCR DNA, GE Healthcare, Chicago, IL, USA). The ligation was performed with T4 DNA ligase (Invitrogen, Carlsbad, CA, USA) for 16 h at 4 °C. After this time, pExu:*hsp65* construction was established by transformation into *E. coli* Top10 by electroporation (1800 V, 200 Ω resistance and 25 μF capacitance pulse in a 0.2 cm cuvette), using the Gene Pulser Xcell™ Electroporation System (Bio-Rad, Richmond, CA, USA). The transformants were plated in a solid medium supplemented with 500 μg/mL of erythromycin to select resistant colonies.

Plasmids from *E. coli* (pExu:*hsp65*) were isolated by alkaline lysis as described by Green and Sambrook [51], and this construction was confirmed by enzymatic digestion (NotI and BamHI). The gene expression was confirmed using a Chinese hamster ovarian cell line [Flp-In™-CHO (Invitrogen, Carlsbad, CA, USA)] (CRL 12023) from ATCC and Lipofectamine 2000 (Invitrogen, Carlsbad, CA, USA) as described by Coelho-Rocha et al. [52]. Briefly, eukaryotic cells were transfected with 4 μg of sterile pExu:*gfp* (positive control); pExu:*hsp65*; pExu:empty [49]; or any plasmid for the negative control. The immunohistochemical reaction checked the eukaryotic cells’ protein expression. Thereby, forty-eight hours post-transfection, the transfected eukaryotic cells were fixed with 4% (*w*/*v*) of paraformaldehyde (Sigma-Aldrich, St. Louis, MO, USA) in phosphate-buffered saline (PBS, 0.1 M) for 15 min and permeabilized with 0.1% (*w*/*v*) Triton ×100 (Vetec, Rio de Janeiro, Brazil) in PBS for 10 min. The cells were incubated for one hour with monoclonal anti-Hsp65 (D17J) (Invitrogen, Carlsbad, CA, USA), diluted 1/50 in 1% bovine serum albumin (BSA; Sigma-Aldrich, St. Louis, MO, USA) at room temperature. The cells were washed with 0.1 M PBS three times and then incubated with the secondary antibody, goat anti-mouse IgG (H + L) Alexa Fluor^®^ 488 (Life Technologies, Carlsbad, CA, USA, 4 µg/mL, diluted 1/500 in PBS/BSA 1%) for 1 h in reduced light conditions. At the same time, 4,6′-diamidino-2-phenylindole (DAPI, Invitrogen, Carlsbad, CA, USA, 2 μg/mL, diluted 1/300) was used for cell nucleus staining. Samples were mounted and the images were captured using a Zeiss LSM 510 META inverted confocal laser-scanning microscope and collected and analyzed using Zeiss LSM Image Browser software. Duplicate transfection assays were performed.

The pExu:*hsp65* functional plasmid was stabilized into *L. delbrueckii* CIDCA133 to develop the recombinant strain, *L. delbrueckii* CIDCA133 (pExu:*hsp65*). To this end, electrocompetent *L. delbrueckii* were transformed by electroporation [2500 V, 200 Ω resistance, and 25 μF capacitance pulse in a 0.2 cm cuvette, Gene Pulser Xcell™ Electroporation Systems (Bio-Rad, Richmond, CA, USA)] with 1 μg/mL of pExu:*hsp65* plasmid. The transformants were plated on MRS (Kasvi, São José dos Pinhais, Brazil) agar (1.5%, Acumedia^®^, Indaiatuba, Brazil) plates supplemented with 2.5 μg/mL of erythromycin for 48 h at 37 °C. Plasmids from *L. delbrueckii* CIDCA 133 (pExu:empty and pExu:*hsp65*) were isolated by glass beads (Sigma Aldrich, St. Louis, MO, USA) using Precellys 24 Homogenizer^®^ (1 cycle, 6500 rpm, 15 s) (Bertin Technologies, Montigny Le Bretonneux, France). After that, the alkaline lysis protocol and enzymatic digestion were carried out.

### 2.3. Dairy Formulation

Fifty microliters of bacteria culture were inoculated into 15 mL of milk medium (12% skimmed milk (*w*/*v*) (Itambé, Belo Horizonte, Brazil), 2% glucose (*w*/*v*) (Labsynth, São Paulo, Brazil), 1.2% (*w*/*v*) yeast extract (Kasvi, São José dos Pinhais, Brazil) with 2.5 µg/mL of erythromycin) for 16 h at 37 °C without shaking to prepare the fermented milk. After that, the culture was diluted 100 times in sterilized milk medium and then administered ad libitum to the mice. To avoid bacterial decantation and clogging, this process was done every 12 h. This beverage and standard chow diet were administered ad libitum for 13 days. The colony-forming units (CFUs) (viable bacteria) administrated to the mice were analyzed by the pour plate method (MRS agar medium with erythromycin, when required), after incubation for 16 h at 37 °C. The CFUs calculated were 3.5 × 10^6^ CFU/mL for *L. delbrueckii* (pExu:*hsp65*), and 5 × 10^6^ for *L. delbrueckii* (pExu:empty).

### 2.4. Mouse Handling and Experimental Design

Conventional BALB/c male, 6 weeks old, weighing 21–24 g, were provided by the animal facility of the Centro de Bioterismo (CEBIO) of the Institute of Biological Sciences, at the Federal University of Minas Gerais (Belo Horizonte, Minas Gerais, Brazil). The animals were kept in polycarbonate open boxes under controlled conditions: temperature around 22 ± 2 °C, 12 h light/dark cycle, humidity of 55 ± 10%, water 24 h before experiments, and standard chow diet available *ad libitum*. All procedures were done in compliance with the Brazilian Society of Sciences in Laboratory Animals (SBCAL) guidelines and were approved by the Animal Experimentation Ethics Committee (CEUA-UFMG, Protocol n° 66/2019, 27 May 2019). The protocol complied with the guidelines recommended by the Institute of Laboratory Animal Resources for the care and use of laboratory animals.

Mice were randomly divided into four different experimental groups (*n* = 8/group): negative control (CTL); CIDCA 133 (pExu:empty), hereafter called rCIDCA 133; positive control of mucositis (MUC); and CIDCA 133 (pExu:*hsp65*), hereafter rCIDCA133:Hsp65. Animals were orally fed daily either with non-fermented milk supplemented with erythromycin (2.5 μg/mL) (CTL and MUC) or with fermented milk by recombinant strains rCIDCA 133 or rCIDCA133:Hsp65 containing erythromycin (2.5 μg/mL) over 13 days. On the 10th day, all groups except the CTL group received a single intraperitoneal (i.p) injection of 5-FU (300 mg/kg) (Fauldfluor^®^, Libbs, São Paulo, Brazil) to induce mucositis, following the same protocol of our previous report [17]. The control group (CTL) received saline solution (i.p). Seventy-two hours after administration, all animals were anesthetized with a ketamine (80 mg/kg) and xylazine (16 mg/kg) mixture (Agener União, Embu-Guaçu, Brazil) and euthanized. The ileum section of the small intestine and the blood of mice were collected for analysis. Furthermore, bodyweight and milk and feed intake were assessed daily.

### 2.5. Intestinal Permeability Evaluation

The intestinal permeability (IP) can assess gut mucosa integrity. A 0.1 mL (18.5 MBq) sample of diethylenetriamine penta-acetic acid (DTPA) solution labeled with technetium-99m (99mTc-DTPA) was administrated by gavage to the mice on the 13th day of treatment, as previously reported [53]. Four hours later, the mice were anesthetized as previously described. Approximately 200 µL blood was collected, weighed, and the radioactivity level in the blood was determined in an automated gamma counter (PerkinElmer Wallac Wizard 1470-020 Gamma Counter, Waltham, MA, USA). A standard dosage containing the same injected amount was counted simultaneously in a separate tube, which was defined as 100% radioactivity. The results were expressed as the percentage of injected dose per gram (%ID/g) of blood: counts per min = (cpm of blood/cpm of the administered dose) × 100. where cpm represents the count of radioactivity per minute.

### 2.6. Bacterial Translocation Study

Bacterial translocation (BT) was evaluated using radiolabeled *E. coli*, as previously reported [54]. Briefly, a culture of *E. coli* ATCC^®^ 10536™ was grown overnight in soybean casein medium agar. Then the grown cells were transferred to 10 mL of sterile 0.9% (*w*/*v*) NaCl solution. Two milliliters of bacterial solution (10^8^ CFU/mL) were incubated with 1 mL of stannous chloride solution (580 mM, pH 7.0) at 37 °C for 10 min. After incubation, 99mTc (37.0 MBq) was added, and the mixture was kept at 37 °C for 10 min. The tubes were then centrifuged at 3000× *g* for 25 min, and 100 mL of either supernatant and resuspended precipitate in saline were used to measure the radioactivity using a dose calibrator (Capintec CRCw-15R Dose Calibrator; CAPINTEC, Inc., Ramsey, NJ, USA). This procedure was done in triplicate. The percent of 99mTc incorporated into the bacterial cells was determined using the following equation: % labeled bacteria = (cpm of precipitate/(cpm of precipitate + cpm of supernatant)) × 100. Then, the suspension of 99mTc—*E. coli* (10^8^CFU/mL) was used for the bacterial translocation study. To this end, 0.1 mL (1.8 MBq) of 99mTc—*E. coli* were administrated by gavage to all groups on the 13th day, 4 h before euthanasia. After this time, the animals were i.p. anesthetized as described before, the blood sample was collected, and then mice were sacrificed by cervical dislocation. Following the mesenteric lymph nodes (MLNs), blood, liver, spleen, heart, kidneys, and lungs were removed, weighed, and the level of radioactivity present in each organ was measured using an automated gamma counter (PerkinElmer Wallac Wizard 1470-020 Gamma Counter; PerkinElmer, Waltham, MA, USA). The results were expressed as counts per minute (cpm)/g of tissue.

### 2.7. Histological, Morphological, and Goblet Cell Analyses

Histological assays were done to analyze the architecture of the intestinal mucosa. After euthanasia, the small intestine was removed, and its length was measured. Then the ileum section was washed with PBS 0.1 M, rolled up, and placed into a histological cassette and immersed in a 10% neutral buffered formalin (NBF) solution [100 mL formaldehyde 37–40%, (Synth, São Paulo, Brazil); 4 g/L HPO4Na, (Synth, São Paulo, Brazil); and 6.5 g/L H_2_PO_4_Na (Vetec, Rio de Janeiro, Brazil)] for 24 h. From a paraffin block containing the samples, 4 μm thick slices were cut, mounted on glass slides, and stained with hematoxylin and eosin (H&E) or periodic acid–Schiff (PAS). To measure the depth of crypts, villus height, goblet cell number, and polymorphonuclear cell infiltrate, the selected image (H&E and PAS) was captured by a BX41 optical microscope (Olympus, Tokyo, Japan) and analyzed using ImageJ 1.51j.8 software (NIH, Bethesda, MD, USA). Image acquisition was performed with a 20× magnification objective and 20 villi and 20 crypts in random fields of each mouse, images were analyzed, and the ratio of villus height⁄crypt depth from the intestinal epithelium was found. The study of goblet cells was performed in PAS-stained slides, where 10 field/slides were counted. Histological examination was performed using a blind score based on a system described previously by Soares et al. [55] to obtain the histological score.

### 2.8. Leukocyte Count

Blood samples were collected from the axial plexus, and the total number of leucocytes was measured by an automatic hematological counter (Bio-2900 Vet, Bioeasy, EUA). Results were expressed as the number of leukocytes per μL of the sample.

### 2.9. Enzyme Assay: Intestinal Myeloperoxidase (MPO) and Eosinophil Peroxidase (EPO) Activity

The MPO and EPO enzyme activities properly evaluate the neutrophil and eosinophil infiltration, respectively, in the intestinal mucosa [56]. These enzymes’ activities were assessed in homogenate ileum tissue described previously by De Jesus et al. [17]. Briefly, 100 mg of tissue were homogenized in 1.9 mL PBS 0.1 M (pH 7.4) using a tissue homogenizer (MA1102 model, Marconi, São Paulo, Brazil). The homogenate was centrifuged (3000× *g* for 10 min), then the pellets were subjected to hypotonic lysis (1.5 mL of 0.2% NaCl). The osmolarity was restored with 1.5 mL of NaCl 1.6% solution supplemented with 5% glucose. Then, samples were centrifuged (3000× *g* for 10 min), and the pellet was resuspended in 0.5% hexadecyltrimethylammonium bromide (HTAB, Sigma-Aldrich, St. Louis, MO, USA) in phosphate buffer. The tissue suspension was homogenized, freeze-thawed three times in liquid nitrogen, and centrifuged for 15 min at 3000× *g*. The resulting supernatant was used in the colorimetric assay to measure the EPO and MPO activities. For EPO assessment, 75 μL of supernatant were added to 75 μL of 1.5 mM O-phenylenediamine (OPD, Sigma-Aldrich, St. Louis, MO, USA), diluted in 0.075 mM Tris–HCl and 6.6 mM H_2_O_2_ (Synth) and incubated at 37 °C for 30 min. For MPO quantification, 25 μL of supernatant were added to 25 μL of 1.6 mM 3,3,5,5′-Tetramethylbenzidine (TMB, Sigma-Aldrich, St. Louis, MO, USA) in dimethyl sulfoxide (DMSO, Sigma-Aldrich, St. Louis, MO, USA). After the addition of 100 μL 0.5 mM H_2_O_2_, the solution was incubated at 37 °C for 5 min. Fifty microliters of 1 M H_2_SO_4_ were added to stop both reactions. Absorbance was measured at 492 nm (EPO) and 450 nm (MPO) on a microplate spectrophotometer (Bio-Rad 450 model, Bio-Rad Laboratories, Hercules, CA, USA). Results were expressed as MPO or EPO arbitrary units (based on absorbance)/100 mg of tissue.

### 2.10. Intestinal Secretory IgA (sIgA)

An enzyme-linked immunosorbent assay (ELISA) was used to measure the sIgA levels in the small intestine, as described by Martins et al. [57]. To this end, the bowel contents were removed, weighed, and flushed out (PBS 0.1 M (pH 7.2) supplemented with aprotinin (1 μM, Sigma-Aldrich, St. Louis, MO, USA), leupeptin (25 μM, Sigma-Aldrich, St. Louis, MO, USA), pepstatin (1 μM, Sigma-Aldrich, St. Louis, MO, USA), and phenylmethanesulfonyl fluoride (PMSF, 1 mM, Sigma-Aldrich, St. Louis, MO, USA)). Samples were centrifuged (2000 rpm for 30 min at 4 °C), and the supernatant was collected to find the immunoglobulin dosage. Microtiter plates (Nunc-Immuno Plates, MaxiSorp) were coated with goat anti-mouse IgA antibody (M-8769, Sigma-Aldrich, St. Louis, MO, USA) in coat buffer (1 M Na_2_CO_3_; 0.1 M NaHCO_3_; pH 9.6) for 18 h at 4 °C, washed (0.1 M PBS + 0.05% Tween 20), and blocked (1% albumin in PBS-Tween 20). Afterward, pre-diluted intestinal fluids (1:1000–0.1 M PBS-Tween 20) were incubated for 1 h. The plates were then washed, and a biotin-conjugated anti-mouse IgA antibody (dil. 1:1000, A4789, Sigma-Aldrich, St. Louis, MO, USA) was added and incubated for 1 h. Finally, 100 μL/well of OPD (1 mg/mL) and 0.04% H_2_O_2_ substrates were added and set for 10 min. The reaction was stopped with 20 μL/well of 1 M H_2_SO_4_ (Sigma-Aldrich, St. Louis, MO, USA). The absorbance was determined at 492 nm using a microplate reader (Bio-Rad model 450, Bio-Rad Laboratories, Hercules, CA, USA). The immunoglobulin concentration was determined using a purified mouse IgA standard (M-8769, Sigma-Aldrich, St. Louis, MO, USA). The concentration of sIgA was expressed in μg/mL of intestinal content.

### 2.11. Transmission Electron Microscopy (MET)

An ileum section (1.5 cm) was fixed in modified Karnovsky solution [(2.5% glutaraldehyde, paraformaldehyde 2.5%, in PBS 0.1 M (pH = 7.2)] at 4 °C for 24 h. Then the solution was removed and substituted with PBS 0.1 M (pH = 7.2), the samples were post-fixed/block stained in 2% osmium tetroxide (OsO4) in PBS 0.1 M for 2 h at room temperature, and 2% uranyl acetate in H_2_O at 4 °C was added and incubated overnight. The fixed tissues were desiderated with ethanol (PA, LabSynth, São Paulo, Brazil) and acetone (PA, LabSynth, São Paulo, Brazil) and embedded in epoxy resin. Semi-thin cuts were done (approximately 300 nm) using a glass razor. The specific sections were chosen in these samples, and ultra-thin cuts (approximately 60 nm) using diamond razors were performed. These sections were placed on a Cu screen (300 mesh) contrasted with lead citrate stains. The images were examined under transmission electron microscopy (TEM; Tecnai G2-1, SpiritBiotwin EIF, 120 kV).

### 2.12. RNA Extraction and Real-Time RT-qPCR of Ileum Section

Total RNA was extracted from 30 mg of ileum tissue, previously stored in 500 μL of RNA later solution (Invitrogen, Carlsbad, CA, USA) to preserve the samples, using TRIzol reagent (Invitrogen, Carlsbad, CA, USA) in compliance with the manufacturer’s guidelines. The RNA concentrations were determined spectrophotometrically using a NanoDrop 2000 (Thermo Scientific, Waltham, MA, USA), considering absorbance ratios of 280/260 and 260/230 nm. The quality of RNA samples was evaluated in agarose electrophoresis gel (1.5%). The extracted RNA was treated for 15 min at room temperature with DNAse I (Invitrogen, Carlsbad, CA, USA). After that, the enzyme was deactivated (10 min 65 °C with 25 mM EDTA). One microgram of total RNA was reverse transcribed using MultiScrib reverse transcriptase (Thermo Fisher, Waltham, MA, USA) in compliance with its guidelines, forming complementary deoxyribonucleic acid (cDNA) using the following parameters: 25 °C for 10 min, 37 °C for 120 min, 85 °C for 5 min.

Quantitative reverse transcription PCR (RT-qPCR) was performed using Applied Biosystems Power SYBR Green PCR master mix (Thermo Fisher, Waltham, MA, USA) and gene-specific primers for Tnf, Il1b, Il6, Il12, Il10, Muc2, claudin 1, 2, and 5, junctional adhesion molecule 1 (F11r), zonulin, occludin, Tlr2, Tlr4, nitric oxide synthase 2 (Nos2), and Myd88 (Table 1). Transcripts were all normalized using Actb and Gapdh [58] housekeeping genes. The experimental approach was optimized by adjusting the primers’ concentrations (5, 10, and 15 pmol) to test for optimal specificity and efficiency. The purity of PCR products was verified by melting curves and gel electrophoresis. The PCR cycle parameters were as follows: initial denaturation at 95 °C for 10 min, 95 °C for 15 s, annealing/extension at 60 °C for 1 min, 40 cycles, followed by a dissociation stage for recording the melting curve. Results were shown graphically as fold changes in gene expression, using the means and standard deviations of target gene expression by Hellemans et al. [59]. Data were analyzed according to the relative expression using the 2^–ΔΔCT^ method. Data are representative of two independent experiments.

### 2.13. Statistical Analysis

The Shapiro–Wilk test assessed data normality. Normal data (body weight loss, small intestine length, sIgA levels, MPO activity, leukocyte count, intestinal permeability, bacterial translocation, villus height to crypt depth ratio, histological score, number of goblet cells, RT-qPCR) were evaluated by analysis of variance (ANOVA) followed by the Bonferroni post hoc test (parametric distribution). Non-normal data (non-parametric distribution) (food and milk intake, EPO activity) were evaluated by the Kruskal–Wallis test followed by Dunn’s post hoc test. A Mann–Whitney test was performed to compare food and milk intake before and after mucositis induction. All data were analyzed using GraphPad Prism 7.0 software, and *p* < 0.05 was considered statistically significant.

## 3. Results

### 3.1. Eukaryotic Cells Can Express Hsp65 Protein

Enzymatic digestion of the new construction, pExu:*hsp65*, confirmed the successful cloning of the Hsp65 sequence in the pExu vector. The immunohistochemical images revealed a precise localization of Hsp protein expression in the cytoplasm of transfected eukaryotic cells with the pExu:*hsp65* plasmid (Figure 1). Thus, the confocal assay confirmed plasmid pExu:*hsp65* functionality. The construction of recombinant strain *L. delbrueckii* CIDCA 133 (pExu:*hsp65*) was confirmed by plating on an MRS/erythromycin plate and plasmid extraction.

### 3.2. rCIDCA 133:Hsp65 Prevented Small Intestine Shortening and Decrease in Weight Loss

Intestinal shortening was observed in inflamed animals, the MUC group (~46 ± 1.80 cm), as expected. Statistical difference was observed (*p* < 0.001) when the MUC group was compared with non-treated animals (CTL group) (~54 ± 1.34 cm). It was possible to observe that treatments with either recombinant strains (rCIDCA 133 and rCIDCA 133:Hsp65) were able to attenuate the small intestine shortening (~50 ± 0.93 and ~52.5 ± 1.34 cm, respectively) (*p* < 0.001). However, intestinal shortening prevention was significantly stronger in animals that received the rCIDCA 133:Hsp65 strain, reaching similar levels to the negative control group (Figure 2A).

The time course of the mice’s weight was another parameter that was evaluated. The bodyweight loss of mice of the rCIDCA 133:Hsp65 group was significantly lower (approximately 4%) than those in the MUC group (about 5.5%) (*p* < 0.01) (Figure 2B). No mortality was observed during the experiment.

Total milk and food intake were similar in all analyzed groups before the induction of mucositis (4 mL and 3 g/day/animal, respectively). There were no significant statistical differences between the experimental groups. After induction of intestinal mucositis, the MUC group showed a reduction in food and milk intake (1.063 ± 0.76 g and 1.208 ± 0.07 mL) when compared with the negative control (2.969 ± 0.25 g and 3.531 ± 0.77 mL) (*p* < 0.01). Nevertheless, the administration of rCIDCA 133 (1.469 ± 0.70 g and 2.094 ± 0.87 mL) and rCIDCA 133:Hsp65 (1.531 ± 0.55 g and 2.750 ± 0.90 mL) treatments was not able to modify these parameters (Figure 2C,D).

### 3.3. rCIDCA 133:Hsp65 Treatment Reduced Ileum Inflammatory Infiltrate and Increased sIgA Levels

The activity of myeloperoxidase (MPO) and eosinophil peroxidase (EPO) was measured in ileum cell lysates to verify whether treatment with rCIDCA 133:Hsp65 could reduce the infiltration of inflammatory cells, like neutrophils and eosinophils, in the intestinal mucosal layer. As exposed in Figure 3A,B, the MUC group (positive control) significantly presented an increase in MPO and EPO enzymes activities, thus showing an increase in neutrophil and eosinophil recruitment (2.453 ± 0.17 U/mg for MPO; 1.561 ± 0.07 U/mg for EPO) (*p* < 0.001) due to the inflammatory processes. Animals treated with rCIDCA 133:Hsp65 showed a reduction in the activity of both enzymes (1.623 ± 0.22 U/mg for MPO, (*p* < 0.001); 0.434 ± 0.04 U/mg for EPO, *p* < 0.01).

A significant reduction in the number of blood leukocytes/μL was observed after 5-FU administration in the MUC group (0.775 ± 0.11 cells × 103/μL) when compared to the negative control (CTL) (3.638 ± 0.46 cells × 103/μL) (*p* < 0.001), as expected. However, the reduction of total leukocyte blood rate induced by chemotherapy was minimized after treatment with either rCIDCA 133 (1.440 ± 0.30 cells × 103/μL) (*p* < 0.01) or rCIDCA 133:Hsp65 (1.814 ± 0.60 cells × 103/μL) (*p* < 0.001). No statistical difference was observed between treatments with both recombinant strains (Figure 3C).

Levels of secretory IgA (sIgA) were also investigated. The results showed that the sIgA levels in the intestinal fluid of the MUC group were significantly increased (3517.2 ± 131.8 µg/mL), and a statistical difference was observed when this group was compared with the negative control (2199.4 ± 330.4 µg/mL) (*p* < 0.001). The treatment with rCIDCA 133 showed significantly reduced levels of sIgA (2905.8 ± 242.6 µg/mL); (*p* < 0.001) in the intestinal fluid of the animals that had received chemotherapy. However, mice treated with rCIDCA 133:Hsp65 were able to increase immunoglobulin levels (3988.2 ± 100.5 µg/mL) significantly, higher than in the MUC group (*p* < 0.01) (Figure 3D).

### 3.4. Reduction in Intestinal Permeability after Treatment with rCIDCA 133:Hsp65

Alteration of mucosal permeability is another side effect of 5-FU treatment. The intestinal permeability was evaluated by measuring radioactivity diffusion in the blood following oral administration of 99mTc-DTPA 72 h after the 5-FU injection. Intestinal permeability was significantly increased in the MUC group (0.258 ± 0.024% ID/g). On the other hand, the CTL group showed lower permeability (0.0187 ± 0.008% ID/g) (*p* < 0.001), as expected. Interestingly, the oral administration with rCIDCA 133:Hsp65 promoted a significant reduction in this parameter (0.0363 ± 0.015% ID/g), reaching similar levels to the negative control (Figure 4A). The same effect was observed in mice treated with the rCIDCA 133 strain. However, the levels of radioactivity were higher than in the rCIDCA 133:Hsp65 group (0.0663 ± 0.026% ID/g and 0.0363 ± 0.015% ID/g, respectively).

### 3.5. rCIDCA 133:Hsp65 Reduced Both Mucosal Damage and Degeneration of Goblet Cells in 5-FU-Induced Intestinal Mucositis

Mice inflamed by 5-FU (MUC group) demonstrated significant alteration in mucosal integrity, such as villus shortening, increased crypt depth, and intense inflammatory cell infiltrate in villi and in lamina propria and submucosa, with ulceration, edema, and vacuolization. Figure 4B shows these findings, which were also confirmed by the scoring system (score 3) [55] (Figure 4C). The rCIDCA 133:Hsp65 and rCIDCA 133 treatments were able to alleviate 5-FU-induced intestinal mucosal damage. The histology assays showed a decreased inflammatory infiltrate, and in the crypt depth and villus height, it was significantly restored (Figure 4D,E). Consequently, the villus height to crypt depth ratio was increased, and histological scores were significantly decreased (1 and 2, respectively) (Figure 4C,F), revealing that both recombinant strains can improve the mucosal preservation in the inflamed mice. However, the treatment with the rCIDCA 133:Hsp65 strain was able to potentiate the decrease in inflammation. A significant decline in goblet cells (12.07± 5.16 cell/field) was observed in the MUC group, as expected. Both recombinant strains studied were able to improve the loss of goblet cell numbers. This protection was significantly higher (36.07 ± 6.64 cell/field) in mice orally treated with rCIDCA 133:Hsp65 than in those which received the rCIDCA 133 strain (26.33 ± 1.98 cell/field) (*p* < 0.01) (Figure 5A,B). On the other hand, the down-regulation of the gene expression of Muc2 mucin (0.17 ± 0.05) after 5-FU administration (MUC group) was observed, with this expression being significantly different from the control groups (Muc2: CTL = 1.10 ± 0.14; rCIDCA 133 = 0.46 ± 0.02) (*p* < 0.05). Animals treated with CIDCA 133:Hsp65 showed an up-regulated expression of the Muc2 gene (0.76 ± 0.17) (*p* < 0.05, Figure 5C).

### 3.6. Tight Junctions Exhibit Up-Regulation after Oral Administration with Recombinant Strains

The results of the gene expression of tight junction proteins, such as Cldn1 (0.15 ± 0.02), Cldn2 (0.06 ± 0.07), Cldn5 (0.26 ± 0.09), occludin (0.14 ± 0.06), zonulin (0.35 ± 0.07), and F11r (0.25 ± 0.14), showed an important gene suppression in ileum tissue in the MUC group, as expected. Both recombinant strains studied, rCIDCA 133 and rCIDCA 133:Hsp65, were able to up-regulate the gene expression of Cldn1 (0.68 ± 0.15; 0.36 ± 0.10) Cldn2 (0.32 ± 0.11; 0.67 ± 0.14), occludin (0.40 ± 0.07; 0.43 ± 0.13), zonulin (0.65 ± 0.16; 0.70 ± 0.09), and F11r (0.71 ± 0.12; 1.07 ± 0.08) (*p* < 005). Animal treatment with rCIDCA 133:Hsp65 strains showed up-regulation in the gene expression of Cldn1 and 2, and F11r (Figure 6A–F).

The ultrastructural examination of ileum mucosa displayed a reduction in the number of microvilli and a greater distance between them. It was possible to observe a higher vacuole number, deficiency of occlusion junctions, and ample open space in adherent junctions and desmosomes compared with control mice. The ultrastructural analyses for either recombinant treatment corroborate the findings of relative gene expression showing, at least in part, that these parameters were ameliorated in animals that received both recombinant strains (Figure 6G).

### 3.7. rCIDCA 133:Hsp65 Reduces Gene Expression of Pro-Inflammatory Molecules and Upregulates the IL10 Expression in Ileum of Inflamed Animals

The qPCR results reveal that the relative mRNA expression of Tnf (7.28 ± 0.42), Il1b (2.3 ± 0.49), Il6 (1.47 ± 0.34), and Il12 (3.01 ± 0.52) were up-regulated in animals from the MUC group, contrary to those exhibited in the CTL group: Tnf (1.0 ± 0.74) (*p* < 0.0001), Il1b (1.0 ± 0.14) (*p* < 0.0001), Il6 (1.0 ± 0.10) (*p* < 0.05), Il12 (1.0 ± 0.17) (*p* < 0.0001). Oral treatment with either rCIDCA 133 and rCIDCA 133:Hsp65 was able to suppress Tnf (1.94 ± 0.86; 1.48 ± 0.13, respectively) (*p* < 0.0001), Il1b (1.04 ± 0.26; 0.40 ± 0.11, respectively) (*p* < 0.0001), Il6 (1.05 ± 0.18; 0.22 ± 0.07, respectively) (*p* < 0.05; *p* < 0.0001), and also Il12 (1.07 ± 0.27; 1.56 ± 0.75, respectively) (*p* < 0.0001; *p* < 0.001). It was observed that levels of the anti-inflammatory Il10 cytokine were reduced (0.08 ± 0.03) (*p* < 0.05) after the administration of 5-FU (MUC group) in relation to the CTL group (1.0 ± 0.27) (*p* < 0.0001). Both recombinant strains studied, rCIDCA 133 and rCIDCA 133:Hsp65, were able to up-regulate Il10 (0.71 ± 0.16; 0.77 ± 0.20, respectively) (*p* < 0.001) expression (Figure 7A–E).

### 3.8. Treatment with Recombinant Strains of Lactobacillus CIDCA 133 Reduces MYD88, NOS2, TLR2, and TLR4 Gene Expression

The MUC group showed a significant increase in mRNA expression of Myd88 (4.16 ± 2.26), Nos2 (2.52 ± 1.090), Tlr2 (2.32 ± 1.08), and Tlr4 (3.26 ± 1.58), when compared with the CTL group: Myd88 (1.00 ± 0.14) (*p* < 0.001), Nos2 (1.0 ± 0.26) (*p* < 0.001), Tlr2 (1.0 ± 0.79) (*p* < 0.01), and Tlr4 (1.0 ± 0.40) (*p* < 0.001). The treatment with the recombinant strains (rCIDCA 133 and rCIDCA 133:Hsp65) resulted in decreased Myd88 (0.65 ± 0.43; 0.03 ± 0.02) (*p* < 0.001; *p* < 0.0001), Nos2 (0.34 ± 0.14; 0.13 ± 0.02) (*p* < 0.0001), Tlr2 (0.77 ± 0.20; 0.08 ± 0.11) (*p* < 0.05; *p* < 0.0001), and Tlr4 (0.60 ± 0.37; 0.03 ± 0.01) (*p* < 0.0001) gene expression. There were no statistical differences when these two recombinant groups were compared (Figure 7F–I).

### 3.9. rCIDCA 133:Hsp65 Was Not Able to Reduce the Bacterial Translocation

Bacterial translocation was evaluated 72 h after the induction of mucositis (Figure 8). Physiological levels of radiolabeled bacteria were detected in all organs, and blood was analyzed from the control group (CTL). This pattern was maintained in animals from the rCIDCA 133 group, except for liver and mesenteric lymph nodes, which were significantly different (*p* < 0.05). Increased amounts of 99mTc–*E. coli* in the blood and all organs were observed in the 5-FU group, as expected. On the other hand, when organs of the rCIDCA: Hsp65 group were analyzed, it was possible to observe a high increment in radiolabeled bacteria. Only heart and kidney showed bacterial translocation similar to physiological levels (*p* < 0.05).

## 4. Discussion

The Hsp65 protein has been intensely studied in different experimental models of inflammation. Studies were performed in a murine model of allergic airway inflammation and hyperresponsiveness to ovalbumin (OVA) (mice sensitized with OVA by i.p. injection and then challenged with OVA by inhalation) and reported that the intramuscular administration of Hsp65 from *M. leprae* before sensitization and challenge was able to prevent the development of these diseases. They reported reduced production of Il4 and Il5. They reported increased Il10 and interferon-gamma (Ifng) in bronchoalveolar lavage fluid [63], with this attenuation effect being caused by Hsp65, acting on the modulation of dendritic cell function, as well as CD4^+^ Th1 cytokine production [64]. Other reports using oral pretreatment with recombinant *L. lactis*, which produces and secretes the Hsp65 protein (*L. lactis* (XIES:hsp65)) [48], showed promissory results: (i) an immunoregulatory effect exhibited by the reduction of pro-inflammatory cytokines (Ifng, Il6 and Tnf), as well as an increase in Il10 in colonic tissues and the expansion of CD4^+^Foxp3^+^ and CD4^+^LAP^+^ regulatory T cells (Treg) in spleen and mesenteric lymph nodes, were able to prevent the Dextran Sulfate Sodium (DSS)-induced colitis in C57BL/6 mice [25] and (ii) prevention of the development of experimental autoimmune encephalomyelitis (EAE) in C57BL/6 mice by reduced inflammatory cell infiltrate and the absence of injury signs in the spinal cord with reduced Il17 and increased levels of Il10 in mesenteric lymph nodes and spleen cell cultures, and also an increase in natural and inducible regulatory T (Treg) cells [26]. Thus, all these findings indicate that Hsp65 from *M. leprae* has a potential therapeutic effect in many inflammation mouse models, and similar effects are shown in this study.

Consistent with this, we decided to develop a DNA vaccine using the pExu vector [49] for the host intestinal cells’ local production of Hsp65 protein. After confirming the pExu:*hsp65* vector’s functionality in eukaryotic cells, the plasmid was transformed in the *L. delbrueckii* CIDCA 133 strain. This strain was isolated from raw cow milk [65] and has probiotic characteristics described as high resistance to acid pH, bile salt, and entero-hemorrhagic microorganisms [65,66,67]. It can decrease harmful bacterial enzymatic activities [68] and resist antimicrobial peptides derived from enterocytes and human β-defensins [69,70]. This strain’s capacity to stimulate phagocytosis by the induction of reactive oxygen species (ROS and NO) and promote the expression of surface markers related to antigen presentation in the in vitro test was described [71]. All these highlighted characteristics have encouraged us to test this strain in the mucositis inflammation model.

The protective effect of fermented milk by wild-type *Lactobacillus delbrueckii* subsp. *lactis* CIDCA 133 strain was previously tested for the first time by our research group in a mucositis mouse model [17]. In this experiment, mice received fermented milk for 13 days, and on the 10th day, they received the 5-FU drug to induce the mucositis. Animals treated with this strain presented a reduction in intestinal permeability values after the administration of 99mTc-DTPA by gavage. These animals also exhibited a preserved villus/crypt ratio, consequently showing a preserved epithelial architecture, with a significant amount of goblet cells and a reduction in inflammatory infiltrate. Altogether, this has shown the beneficial effect of this probiotic strain in intestinal inflammation, specifically in intestinal mucositis.

Therefore, combining the excellent reported characteristics of this probiotic strain with the benefits of the Hsp65 protein, we investigate the oral treatment capacity with recombinant *L. delbrueckii* CIDCA 133 (pExu:*hsp65*)-fermented milk in a mucositis mouse model.

In this work, mice who received treatment for 13 days with recombinant *L. delbrueckii* CIDCA 133 (pExu:*hsp65*)-fermented milk could avoid the intestinal shortening, reaching a length similar to the negative control. This parameter is very relevant to correlate the nutrient (food and liquid) uptake and the weight loss percentage, and consequently, the state of the animals’ malnutrition. Thus, it was possible to attribute the lower weight loss to the group treated with *L. delbrueckii* CIDCA 133 (pExu:*hsp65*). According to a previous report, this effect was due to this bacteria’s probiotic effect on the bowel’s length [17].

Severe epithelial damage to the intestinal mucosa, especially in the jejunum and ileum sections, was related to mucositis pathobiology. Epithelial architecture and integrity damage with active migration of polymorphonuclear leukocytes (PMNs) stimulated by increased adhesion molecules from nuclear factor kappa B (NF-kB), through the intestinal villus and crypts, are typical signs of mucositis 1. Thus, neutrophil infiltration has been described in several studies that assessed mucositis [17,53,72,73,74,75]. The beneficial effect of fermented milk by *L. delbrueckii* rCIDCA 133:Hsp65 on attenuating the intestinal inflammation was demonstrated by the decrease in MPO activity, the histological score with lower inflammatory infiltrate, and higher villus/crypt ratio. Additionally, the restoration of the epithelium’s architecture was shown in this work, and the low histological score of these animals was the same as the negative control. Altogether, this denotes the beneficial effect of this treatment.

The beneficial effect of the treatment with *L. delbrueckii* rCIDCA 133:Hsp65 could also be observed in the increased number of goblet cells and the relative expression of the Muc2 gene. Mucins are secreted by the goblet cells and create a layer that protects the epithelium against bacterial penetration [76]. Thus, the higher number of these cells and also the level of expression of Muc2compared to the MUC group reinforce the positive effect of the recombinant treatment.

Secretory IgA is a primary immunologic component of the mucosal surface’s extrinsic protective mechanism [77]. Our group’s previous reports observed increased sIgA levels in the small bowel of inflamed groups [17,22,78]. This event is related to the intestinal inflammatory process of a host’s defense mechanism [79]. However, other authors report a decrease in sIgA levels in inflamed groups [53,80], attributed to malnutrition generated by 5-FU reducing the sIgA responses and Gut-Associated Lymphoid Tissue (GALT) lymphocyte numbers. Following our previous reports, high levels of sIgA were observed in animals that did not receive any recombinant CIDCA 133 strain (MUC group), even more than in animals that received rCIDCA133:Hsp65.

On the other hand, the rCIDCA133 group manifested decreasing levels of this immunoglobulin. This immunoglobulin plays an essential role in different immune system functions and existing IgA antibodies with high and low affinity. Those with high affinity protect intestinal mucosal surfaces against invasion by pathogenic microorganisms, while low-affinity antibodies confine commensal bacteria to the intestinal lumen [81].

In healthy conditions, a small number of bacteria from the gut microbiota can translocate, and are killed during this passage or in the MLNs, a phenomenon that contributes to the maturation and maintenance of a competent gastrointestinal immune system [82,83,84]. In agreement with this affirmation, the CTL group showed a basal count of radioactivity in the blood and organs. Interestingly, animals treated with rCIDCA 133 only showed higher values of uptake of 99mTc-*E. coli* in MLNs and liver and this event could be explained by gut injury or barrier failure contributing to the translocation of bacteria from the gut as MLNs are the first structures to receive the gut bacteria which, by portal circulation, reach the liver. This translocation pattern was demonstrated in other experimental models, which showed that viable bacteria could be found in the portal circulation, in high amounts, even before their appearance in intestinal lymph [85].

As the damage could be solved partially by rCIDCA 133, the other organs’ translocation was the same as in the negative control, which presented the basal count of radioactivity.

Extremely high values of radioactivity were observed in the liver of animals that received rCIDCA133:Hsp65. These results could be attributed to the existence of the gut–liver axis, which is one of the most significant relations between gut microbiota and extra-intestinal organs, representing an extremely close functional and bidirectional communication between these two organs [86]. Damage in the intestinal barrier automatically exposes the liver to bacterial components, such as pathogen-associated molecular patterns (PAMPs) and damage-associated molecular patterns (DAMPs), which could damage the organ by induction of an immune response, with the release of pro-inflammatory cytokines [87]. As the Hsp65 molecule has a dual role, inflammatory and regulatory, the liver’s high inflammation due to the closer relationship with the gut was not enough to block the dissemination of 99mTc-*E. coli* to the rest of the body. Additionally, as is often reported, the 5-FU cancer treatment generates a disruption of the indigenous microbiota’s ecological balance and damage of the gut mucosal barrier by all the events involved in mucositis [88,89], which could lead to dysbiosis [90,91,92] with epithelium damage. The enteric venous system to the portal vein and the enteric lymphatic drainage are routes that could contribute to delivering bacteria or bacterial compounds from the gut to the circulation [93]. BT was the only parameter for which animals treated with rCIDCA133:Hsp65 did not show satisfactory results. These results can be explained, at least in part, by the fact that Hsps are closely related to apoptosis and ROS generation in human inflammatory diseases [94,95,96], and it is also known that oxidative stress metabolites can increase epithelial permeability [97]. Even though the treatment with recombinant bacteria has shown a reduction in the permeability, mucosal atrophy, mostly that associated with luminal nutrient deprivation, can be observed in mucositis, has been suggested as a predisposing factor for bacterial translocation [98]. Berg et al. [99] showed that immunosuppression associated with intestinal bacteria overgrowth promoted bacterial translocation in animals with histologically normal bowels. However, further studies must be performed to elucidate if this hypothesis can be applied to this specific mucositis model.

Moreover, two pathways could contribute to gastrointestinal permeability, the transcellular passage of molecules through the enterocytes using channels and membrane pumps, through the apical and basolateral membranes [100,101]. The paracellular route is regulated by tight junction proteins situated between adjacent enterocytes’ apical lateral membranes, holding epithelial cells together. Their conservation and preservation are essential for cellular polarity and the intestinal epithelium’s epithelial barrier function [93,102]. It is also known that tight junctions prevent the transepithelial movement of lipopolysaccharides and other macromolecules and bacteria [103].

The intestinal permeability was studied either by blood radioactivity after the oral intake of 99mTc-DTPA or by relative mRNA expression of tight junction proteins. It was shown that these two approaches are related because an improvement in this parameter (IP) was observed. The percentage of doses/g of 99mTc-DTPA was significantly lower in animals treated with recombinant strains. Treatment with the rCIDCA133:Hsp65 strain showed similar values to the negative control. Our results are in agreement with other reports where permeability was measured after probiotic treatment [17,53]. All the tight junction proteins studied, except claudin 5, significantly increased the mRNA expression with recombinant treatment. When the mRNA expression of tight junctions in the ileum section was analyzed, an increase was observed. However, the gene expression of claudin 5 did not show increased mRNA expression.

However, the group treated with the milk fermented by rCIDCA133:Hsp65 showed a higher expression in F11r, claudin 1, and claudin 2, although occludin and zonulin did not show a difference between these two treatments, demonstrating that both recombinants strains were able to increase these relative tight junction expressions. The qPCR results were reinforced by transmission electronic microscopy analyses showing improved organization of zonula occludens, zonula adherents, desmosomes, and gap junctions in animals treated with the recombinant strains when compared to the ones of the positive control (MUC group). Altogether, these results show the relevant findings related to the intestinal permeability improvement after the fermented milk administration. These results agree with other reports showing the importance of probiotics in tight junction expression and, consequently, intestinal permeability [104,105,106].

There is a closer relation between intestinal tight junctions and cytokines’ role under pathophysiological conditions. The disfunction of tight junctions mediated by cytokines leads to immune activation and tissue inflammation [107,108].

The exposition to cytotoxic therapy, such as 5-FU, leads to direct damage to cellular DNA, generating cell injury and death in both basal epithelium and submucosal cells and, consequently, increasing reactive oxygen species (ROS) [109]. They cause more severe damage to the involved tissues with macrophage stimulation and also activate the cascade of inflammatory pathways and transcription factors, such as NFKB 1 [110], which facilitates both the up-regulation of Il1b [111] and also the synthesis of pro-inflammatory molecules such as Tnf, Il6, cyclooxygenase-2 (Ptgs2), and Il12, among others [112,113].

In this study, the anti-inflammatory effect provided by both recombinant strains was shown. Milk fermented by both recombinant strains can attenuate the relative expression of the pro-inflammatory cytokines studied (Il1b, Il6, Il12, and Tnf), evidencing the anti-inflammatory effect that these strains provide. The milk fermented by rCIDCA 133:Hsp65 was more effective in decreasing Il1b and Il6, leading to similar levels in the non-inflamed animals. The importance of these findings is due to the significant role in inflammation of both cytokines [114], participating as mediators in the course of intestinal apoptosis after 5-FU chemotherapy [115], and also causing an increase in intestinal permeability. Il1b and Tnf participate in amplifying the severity of chemotherapy-induced intestinal mucositis [1]. Tnf is relevant because it induces the activation and the recruitment of PMN cells, causing damage and intestinal barrier dysfunction, increasing the intestinal permeability, and, consequently, levels of pathogenic bacteria from the intestinal lumen [116,117]. Il6 production is a hallmark of many human chronic inflammatory states, including mucositis due to its pro-inflammatory properties [118,119]. It has a high level of participation in the inflammatory Tlr4-mediated pathway [120]. The basal levels of Tlr4 are fundamental to control physiological states and intestinal homeostasis [121]. Tlr4 knockout mice showed the importance of this pattern recognition receptor (PRR) in signaling through Myd88 of bacterial translocation [122]. This information could explain our results regarding the high level of translocation observed in animals treated with rCIDCA:Hsp65 since they presented much lower levels, even lower than the negative control, of Tlr4 and also Myd88 mRNA expression. However, further experiments should be performed to elucidate the mechanisms of this complex signaling pathway.

As 5-FU generates a dysbiosis in the bowel, disruption of the mucosal barrier upon injury to intestinal epithelial cells causes the exhibition of Tlr ligands by commensal bacteria to Tlr expressed in many bowel cells [123]. Thus, due to the significant role of the microbiota in intestinal inflammation, the active participation of Tlr in bacterial product recognition and its importance in inflammation induction encourages us to test the in vivo effect of Tlr ligation by commensal-derived products. Other essential intracellular adapters, such as Myd88 and Nod2, were also investigated. Tlrs (except Tlr3) stimulate the cells through Myd88, which mediates the early immune response to pathogens, leading to Nfκb translocation to the nucleus and, consequently, gene expression of encoded pro-inflammatory cytokines and chemotactic cytokines (chemokines) [124,125]. The microbial product also can be recognized by a family of intracellular signaling proteins, called Nod. They can detect microbial ligands in the cytosol. These cytosolic proteins signal Nfkb and MAPK, followed by the induction of numerous genes involved in the inflammatory process, thus triggering host innate immune responses [126,127,128]. Nod2 recognizes muramyl dipeptide on degraded bacterial cell wall peptidoglycan and can therefore respond to invading Gram-negative and Gram-positive bacteria.

Our results follow those of other authors who describe Tlrs as the major sensors of the innate immune system which recognize highly preserved motifs of microorganisms [129], and are related to the activation of signaling cascades, such as Nfkb, associated with inflammatory responses through the production of pro-inflammatory cytokines [130,131]. Additionally, an increase in the Tlr2 and Tlr4 levels was observed in animals that received 5-FU, and the administration of probiotics was able to reverse these increases [132], as we could demonstrate. This analysis shows, at least, a potential relationship between these parameters. However, mRNA expression cannot affirm that different circulant cytokines’ different concentrations will be found due to the possible post-translational modification.

The Il10 cytokine is an important immunoregulatory molecule required to maintain immune homeostasis in the gut and contributes to declining inflammatory responses by down-regulating pro-inflammatory cytokine production at the site of tissue damage [133]. It is the most well-researched cytokine in inflammatory bowel disease (IBD), and its active form is secreted by different types of immune cells, such as Treg, monocytes, and macrophages [134,135]. Studies support the hypothesis that Il10 can attenuate Tnf receptor expression [136,137], which can be related to the results found in this study. In contrast, treatments with both recombinant strains were able to highly improve Il10 expression, with a reduction of Tnf, Il1b, and Il6 expression levels. These results are also supported by histological, MPO, and EPO analyses.

The results together show that animals undergoing induction chemotherapy with 5-FU and pretreated with *L. delbrueckii* CIDCA 133 (pExu:*hsp65*) experienced a reduced magnitude of damage to the mucosal architecture of the small intestine and decreased intestinal permeability, highlighting the importance that this probiotic recombinant strain has in reestablishing the number of goblet cells. Thus, the recombinant probiotic *L. delbrueckii* CIDCA 133 (pExu:*hsp65*) consumption could be an excellent alternative to ameliorating the intestinal damage caused by 5-FU in a mouse model.

## Figures and Tables

**Figure 1 microorganisms-09-00107-f001:**
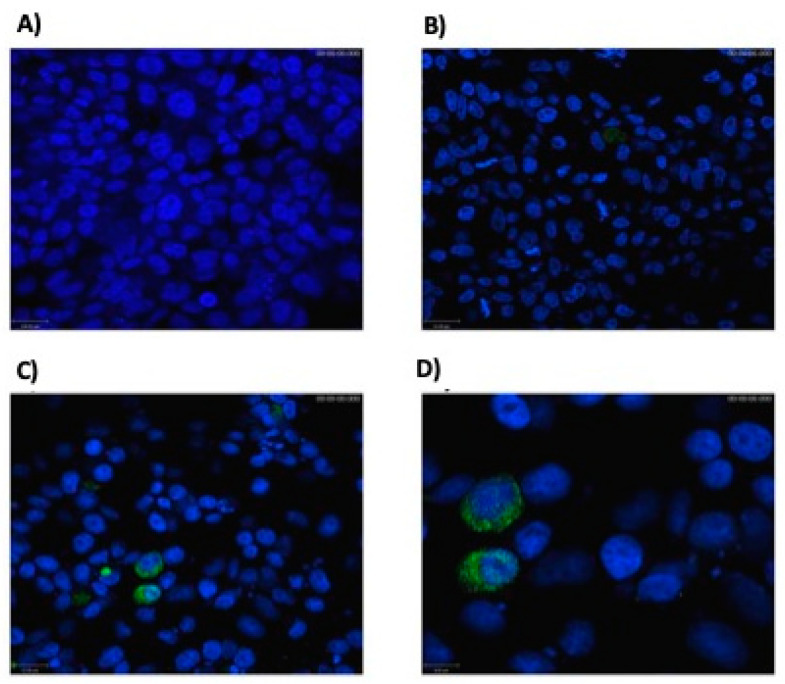
Hsp65 expression in transfected eukaryotic cells. Confocal microscopy: (**A**) negative control: non-transfected Chinese Hamster Ovarian cell line (CHO) cells; (**B**) negative control: non-transfected eukaryotic cells labeled with primary (Mab anti-Hsp65) and secondary (goat anti-mouse IgG (H + L)) antibodies labeled with Alexa 488. (**C**,**D**) CHO cells labeled with primary (Mb_HSP65) and secondary (goat anti-mouse IgG (H + L)) antibodies. In green, the Hsp65 protein is expressed in the cytoplasm of eukaryotic cells. 2D images (**A**–**D**) were acquired in both depths (z-stack) using a Zeiss LSM 510 META inverted confocal laser 1358 scanning microscope with 40× or 60× objective.

**Figure 2 microorganisms-09-00107-f002:**
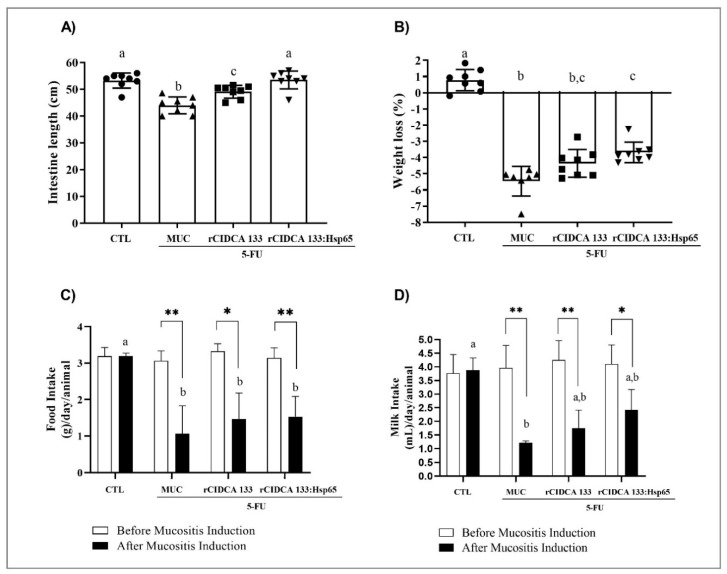
Small intestine length, bodyweight, food and milk intake analysis: (**A**) intestine length, (**B**) bodyweight variation (**C**) food intake, and (**D**) milk intake. (**A**,**B**) ANOVA followed by the Bonferroni post hoc test and (**C**,**D**) Kruskal–Wallis test followed by Dunn’s post hoc test. Different letters (a, b, and c) indicate statistically significant differences between groups (*p* < 0.05). The symbols (*) (**) show a statistically significant difference (*p* < 0.05) (*p* < 0.01), respectively, between rCIDCA 133, MUC (positive control), and rCIDCA 133:Hsp65 groups before and after mucositis induction by an unpaired Student’s *t*-test (**C**,**D**). Geometric symbols show to the number of animals evaluated in each experimental group. • refers to the animals of CTL group, ▲ refers to the animals of MUC group, ■ refers to the animals of rCIDCA 133 group and ▼ refers to the animals of rCIDCA 133:Hsp65 group.

**Figure 3 microorganisms-09-00107-f003:**
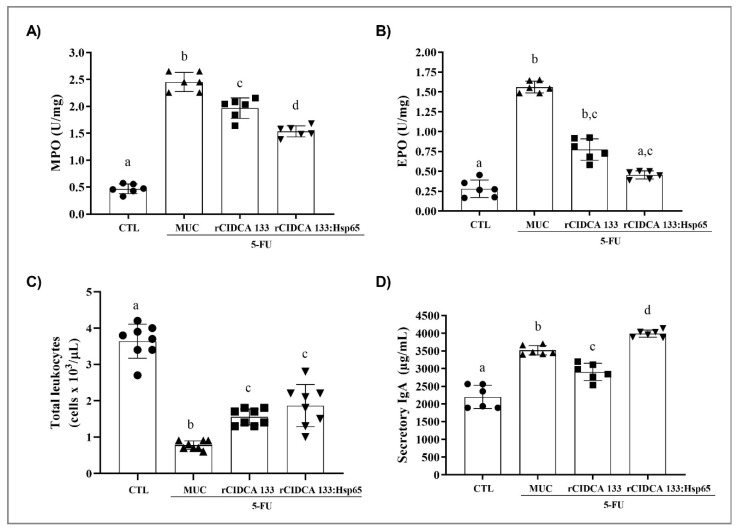
Effect of rCIDCA 133:Hsp65 on inflammatory parameters and epithelial barrier of animals inflamed with 5-fluorouracil (5-FU): (**A**) Intestinal Myeloperoxidase (MPO) and (**B**) Eosinophil Peroxidase (EPO) activity, (**C**) intestinal permeability (%ID/g of 99mTc-DTPA), and (**D**) levels of Intestinal Secretory IgA (sIgA) (μg/mL). Mice received intraperitoneal 5-FU (300 mg/kg) (MUC, rCIDCA 133, and rCIDCA 133:Hsp65 groups) or 0.9% saline solution (CTL group) and were treated with non-fermented milk or recombinant *L. delbrueckii* CIDCA 133-fermented milk (*n* = 6 animals per group). Different letters indicate statistically significant differences (*p* < 0.05) by ANOVA followed by the Bonferroni post hoc test (**A**,**C**,**D**) and Kruskal–Wallis test followed by Dunn’s post hoc test (**B**). Geometric symbols show to the number of animals evaluated in each experimental group. • refers to the animals of CTL group, ▲ refers to the animals of MUC group, ■ refers to the animals of rCIDCA 133 group and ▼ refers to the animals of rCIDCA 133:Hsp65 group.

**Figure 4 microorganisms-09-00107-f004:**
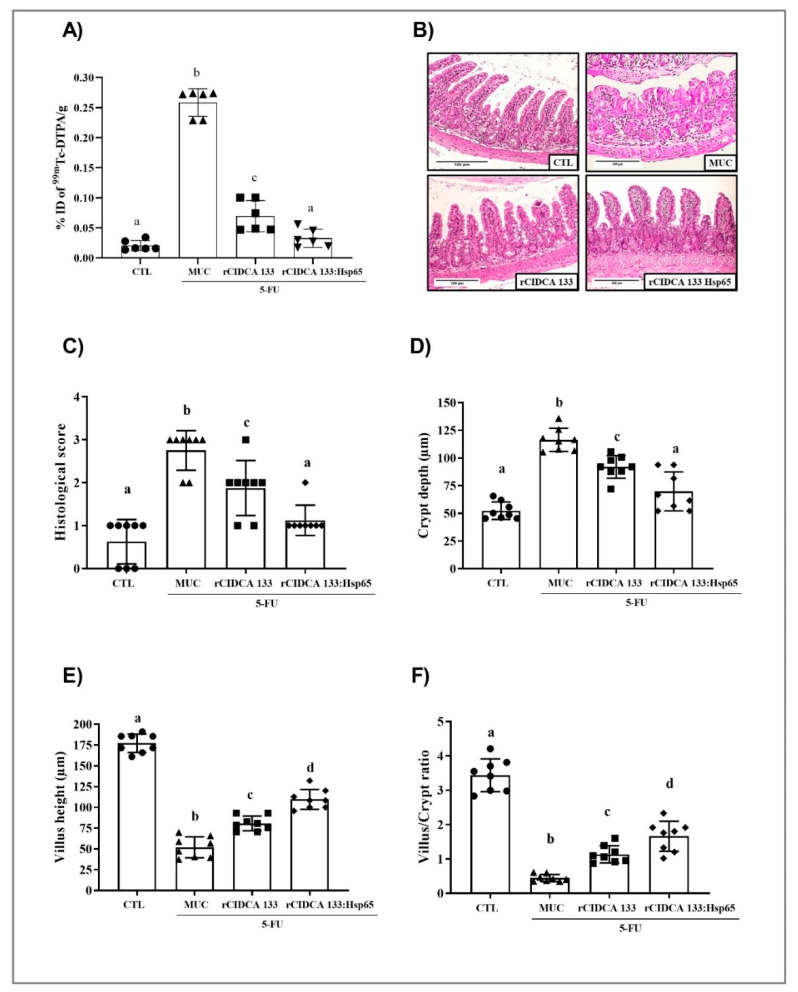
Histopathological and morphometric analysis, intestinal permeability, and evaluation of the relative gene expression of tight junction proteins: (**A**) intestinal permeability (%ID/g of 99mTc-DTPA), (**B**) mucosal histopathology, (**C**) histopathological scores of the ileum of animals (objective: ×20, scale 100 µm), (**D**) morphometrical analysis of crypt depth, (**E**) villus height, (**F**) villus height to crypt depth ratio. Mice received intraperitoneal 5-FU (300 mg/kg) (MUC, rCIDCA 133, and rCIDCA 133:Hsp65 groups) or saline solution (CTL group). They were treated with non-fermented milk supplemented with erythromycin 2.5% (CTL and MUC) or recombinant *L. delbrueckii* CIDCA 133 (rCIDCA133 and rCIDCA133:Hsp65)-fermented milk supplemented with erythromycin 2.5% (*n* = 8 animals per group). Different letters (a–d) indicate statistically significant differences (*p* < 0.05) by ANOVA followed by the Bonferroni post hoc test. Geometric symbols show to the number of animals evaluated in each experimental group. • refers to the animals of CTL group, ▲ refers to the animals of MUC group, ■ refers to the animals of rCIDCA 133 group and ▼ refers to the animals of rCIDCA 133:Hsp65 group.

**Figure 5 microorganisms-09-00107-f005:**
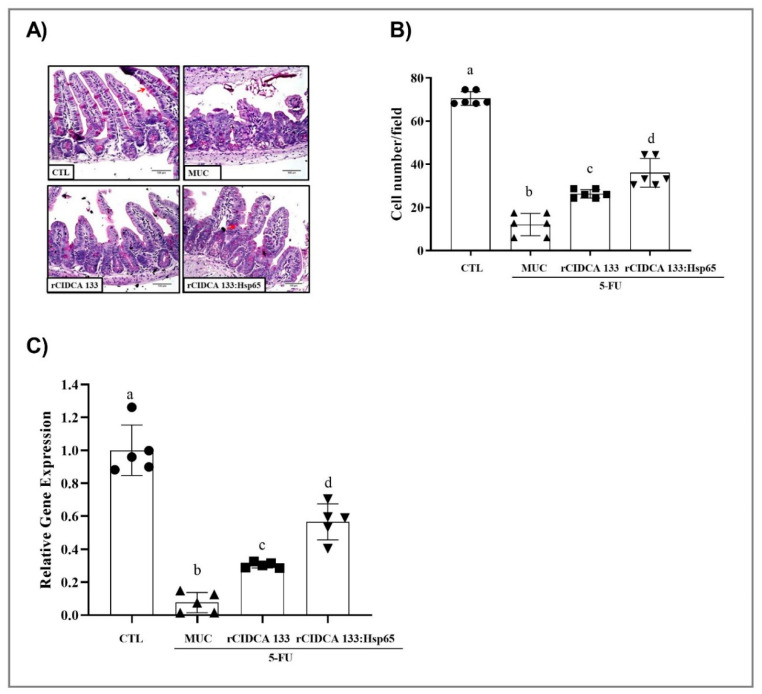
Assessment of goblet cell integrity and evaluation of the relative gene expression of Muc2 mucin: (**A**) representative photomicrographs from ileum section stained with Periodic Acid–Schiff (PAS), the arrows show the goblet cells (objective: ×20, scale 100 µm), (**B**) number of goblet cells/field obtained for experimental groups, and (**C**) level of relative mRNA expression of Muc2 mucin. Mice received intraperitoneal 5-FU (300 mg/kg) (MUC, rCIDCA 133, and rCIDCA 133:Hsp65 groups) or saline solution (CTL group). They were treated with non-fermented milk supplemented with erythromycin 2,5% (CTL and MUC) or recombinant *L. delbrueckii* CIDCA 133 (rCIDCA133 and rCIDCA133:Hsp65)-fermented milk supplemented with erythromycin 2,5% (*n* = 6 animals per group). Different letters (a–d) indicate statistically significant differences (*p* < 0.05) by ANOVA followed by the Bonferroni post hoc test (**B**,**C**). Geometric symbols show to the number of animals evaluated in each experimental group. • refers to the animals of CTL group, ▲ refers to the animals of MUC group, ■ refers to the animals of rCIDCA 133 group and ▼ refers to the animals of rCIDCA 133:Hsp65 group.

**Figure 6 microorganisms-09-00107-f006:**
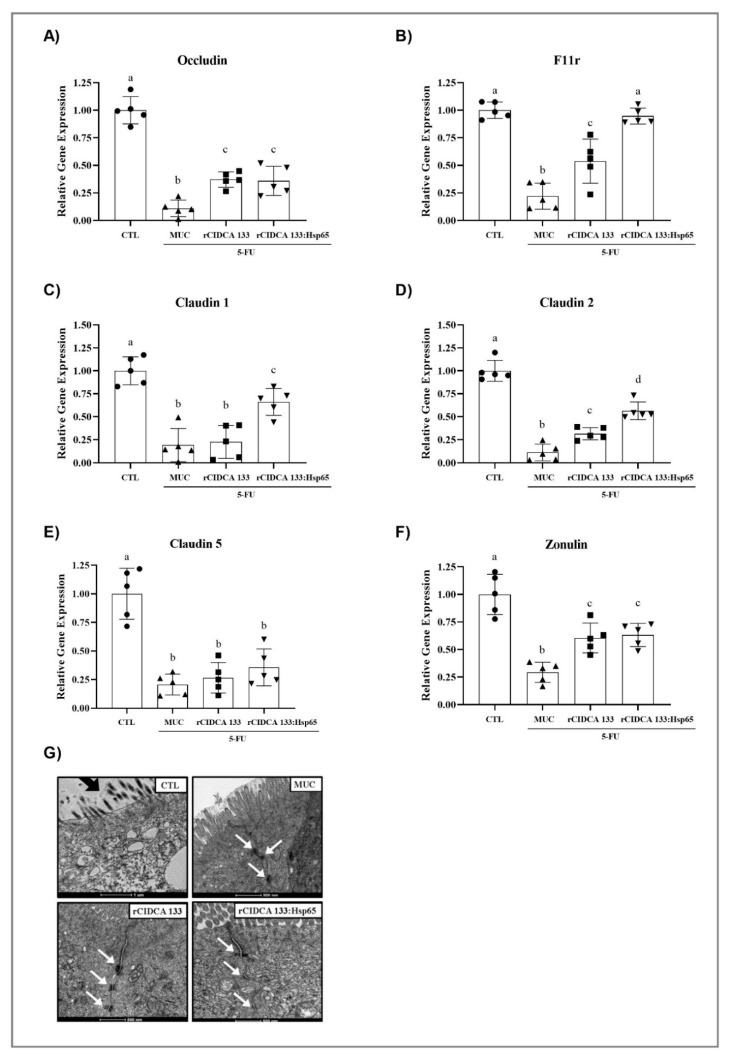
Level of relative mRNA expression of (**A**–**F**) tight junction proteins (occludin, F11r, claudin 1, claudin 2, claudin 5, and zonulin) determined by qPCR from animal ileum. Different letters (a–d) indicate statistically significant differences (*p* < 0.05) by ANOVA followed by the Bonferroni post hoc test (**A**–**F**). Geometric symbols show to the number of animals evaluated in each experimental group. • refers to the animals of CTL group, ▲ refers to the animals of MUC group, ■ refers to the animals of rCIDCA 133 group and ▼ refers to the animals of rCIDCA 133:Hsp65 group. (**G**) Evaluation of cell junctions by transmission electron microscopy: dark arrow highlights the reduction in the number and length of microvilli of the MUC group cells compared to the other groups. The white arrows highlight desmosomes’ presence at the lateral junctions between the epithelial cells in the CTL (scale = 1 µm), MUC, rCIDCA133, and rCIDCA 133:Hsp65 (scale = 500 nm) groups.

**Figure 7 microorganisms-09-00107-f007:**
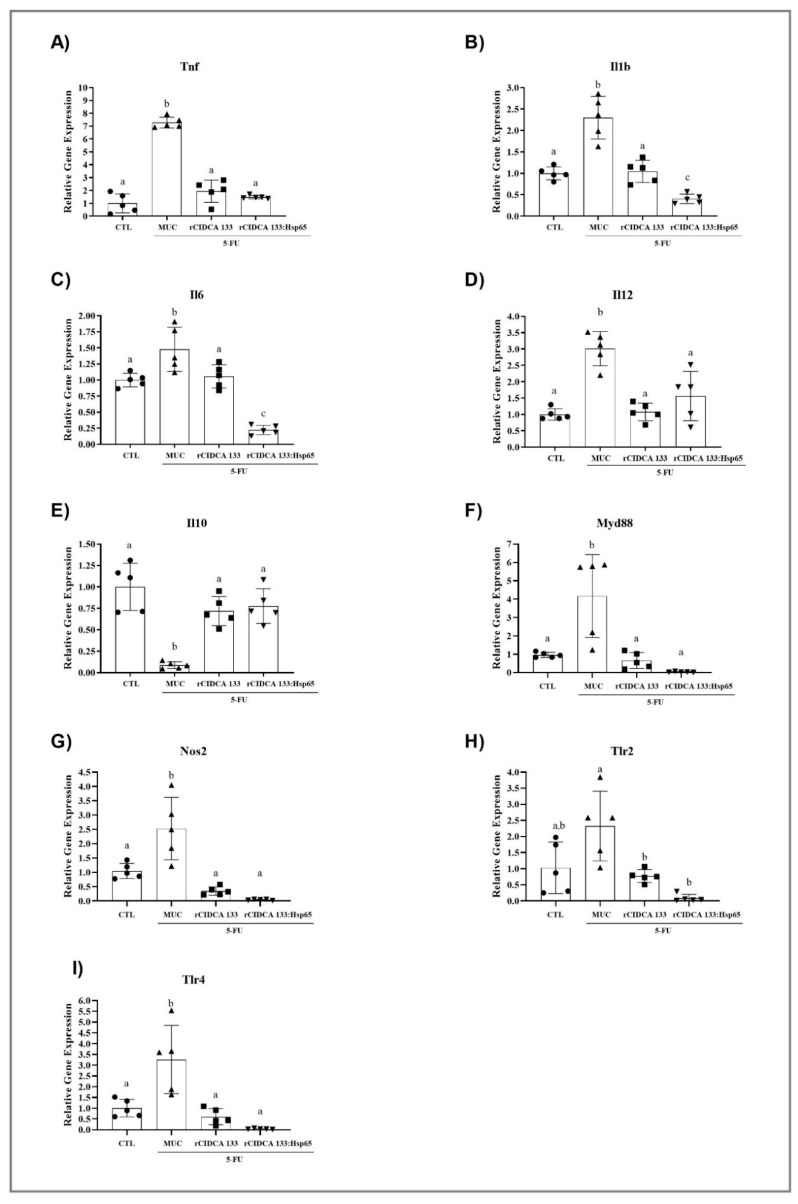
Relative gene expression of (**A**–**I**) Tnf, Il1b, Il6, Il12, Il10, Myd88, Nos2, Toll-like receptors 2 and 4. Mice received intraperitoneal 5-FU (300 mg/kg) (MUC, rCIDCA 133, and rCIDCA 133:Hsp65 groups) or saline solution (CTL group). They were treated with non-fermented milk supplemented with erythromycin 2.5% (CTL and MUC) or recombinant *L. delbrueckii* CIDCA 133 (rCIDCA133 and rCIDCA133:Hsp65)-fermented milk supplemented with erythromycin 2.5% (*n* = 5 animals per group). Different letters indicate statistically significant differences (*p* < 0.05) by ANOVA followed by the Bonferroni post hoc test. Geometric symbols show to the number of animals evaluated in each experimental group. • refers to the animals of CTL group, ▲ refers to the animals of MUC group, ■ refers to the animals of rCIDCA 133 group and ▼ refers to the animals of rCIDCA 133:Hsp65 group.

**Figure 8 microorganisms-09-00107-f008:**
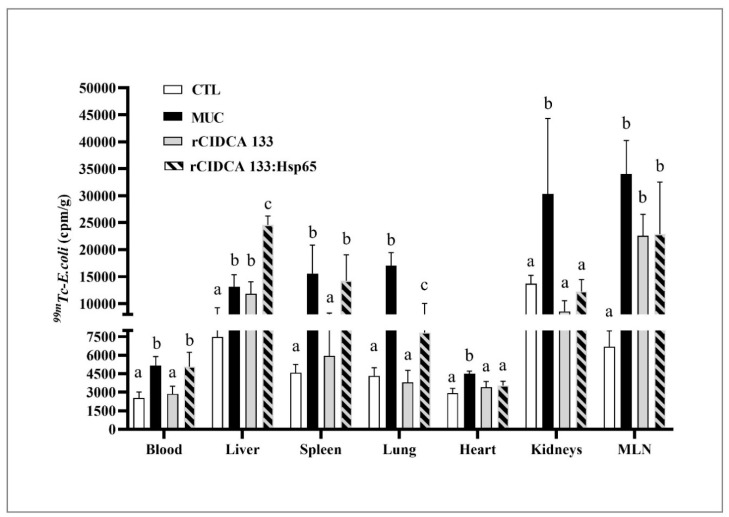
Biodistribution of 99mTc-*Escherichia coli* in animals with intestinal mucositis and treated with rCIDCA 133:Hsp65. cpm, counts per minute; MLN, mesenteric lymph node. Different letters indicate statistically significant differences (*p* < 0.05) by ANOVA followed by the Bonferroni post hoc test.

**Table 1 microorganisms-09-00107-t001:** Primers used in qPCR.

Gene	Foward Primer	Reverse Primer	Reference
Actb	GCTGAGAGGGAAATCGTGCGTG	CCAGGGAGGAAGAGGATGCGG	[60]
Gapdh	TCACCACCATGGAGAAGGC	GCTAAGCAGTTGGTGGTGCA	[58]
Il6	GAGGATACCACTCCCAACAGACC	AAGTGCATCATCGTTGTTCATACA	[58]
Il10	GGTTGCCAAGCCTTATCGGA	ACCTGCTCCACTGCCTTGCT	[58]
Il12p40	GGAAGCACGGCAGCAGAATA	AACTTGAGGGAGAAGTAGGAATGG	[58]
Tnf	ACGTGGAACTGGCAGAAGAG	CTCCTCCACTTGGTGGTTTG	[61]
Il1b	CTCCATGAGCTTTGTACAAGG	TGCTGATGTACCAGTTGGGG	[61]
Muc2	GATGGCACCTACCTCGTTGT	GTCCTGGCACTTGTTGGAAT	[60]
Myd88	ATCGCTGTTCTTGAACCCTCG	CTCACGGTCTAACAAGGCCAG	[62]
Tlr2	ACAATAGAGGGAGACGCCTTT	AGTGTCTGGTAAGGATTTCCCAT	[62]
Tlr4	ATGGCATGGCTTACACCACC	GAGGCCATTTTTGTCTCCACA	[62]
Nos2	CAGCTGGGCTGTACAAACCTT	CATTGGAAGTGAAGCGTTTCG	[58]
Cldn1	TCCTTGCTGAATCTGAACA	AGCCATCCACATCTTCTG	[60]
Cldn2	GTCATCGCCCATCAGAAGAT	ACTGTTGGACAGGGAACCAG	[60]
Cldn5	GCTCTCAGAGTCCGTTGACC	CTGCCCTTTCAGGTTAGCAG	[60]
Occludin	ACTCCTCCAATGGACAAGTG	CCCCACCTGTCGTGTAGTCT	[60]
Zonulin	CCACCTCTGTCCAGCTCTTC	CACCGGAGTGATGGTTTTCT	[60]
F11r	CACCTTCTCATCCAGTGGCATC	CTCCACAGCATCCATGTGTGC	[60]

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
