# Peer review of "Intake of Lactobacillus delbrueckii (pExu:hsp65) Prevents the Inflammation and the Disorganization of the Intestinal Mucosa in a Mouse Model of Mucositis"

_microorganisms, 2021, doi:10.3390/microorganisms9010107_

Round 1

Reviewer 1 Report

I read with great interest the article by Lima Barroso et al exploring the beneficial effect of oral administration of fermented milk by Lactobacillus
30 delbrueckii CIDCA 133 (pExu:hsp65), a recombinant strain, in a model of mucositis. Before acceptance the authors should address the following points:

Introduction: the authors should cite the known effect of Hsp65 and which is the putative mechanisms for Hsp65 acting on mucositis induced by anti-neoplastic

Methods: the authors should explain the reason why they used for assessing intestinal permeability the scintigraphy instead of test with sorbitol or mannitol. 

Moreover, were zonulin levels analyzed?

For inflammation: were cytokine levels analyzed?

Study design: was the intestinal permeability and inflammation assessed even after L. delbrueckii (not hsp65) administration as a negative control? It is widely know that Lactobacilli themselves act reducing permeability inflammation, even in other settings (see and amend doi:10.3390/nu12092674)

RESULTS: The authors should conform figures and graphics style always using colour figures and graphs.

  • For points 3.7 and 3.8 the authors should report p values among values in parenthesis

DISCUSSION

The authors should discuss the results of previous studies on the use of L. delbruecki (not hsp65 recombinant) on intestinal inflammation, permeability and more in general, in available, on human patients on 5-FU therapy.

There are several typos throughout the manuscript that need correction

Author Response

Dear Dr. 

Following, please see the ansewrs regarding your appointments.

Introduction: the authors should cite the known effect of Hsp65 and which is the putative mechanisms for Hsp65 acting on mucositis induced by anti-neoplastic

We add a paragraph regarding hsp65 known effects. Please see below and in the manuscript  between lines 79-87.

HSPs, also known as stress proteins, are considered conserved proteins during evolution, present in all living organisms. These proteins have an essential role as molecular chaperones interacting either with proteins tagged for degradation or with foreign polypeptides aggregation (Saibil et al., 2013).  HSPs are upregulated when cells face stressful situation (Lindquist et al 1998, Tsan & Gao 2009; Coelho & Faria 2012) as an infection disease; presenting immunoregulatory activities (Nishikawa et al., 2008) and also participate in cell survival signaling pathways (Chun et al., 2010). Also, these molecules are considered natural adjuvants since they stimulate the internalization by scavenger receptors and also the presentation of antigens through MHC molecules, inducing the production of chemokines, pro-inflammatory cytokines, as well as the production of nitric oxide by macrophages and dendritic cells (Lehner et al., 2000; More, Breloer & Von Bonin, 2001, Nishikawa et al 2008).

We add some information about the mechanisms possibly related to the beneficial effect of Hsp65. Please see below and in the manuscript between lines 88-101.

It is still not clear the precise role of Hsps in IBDs. Miao and colleagues (2014) correlated the severity of the disease with elevated levels of Hsp70 in human patients with Ulcerative Colitis. In patients with IBD, autoantibodies were detected to Hsp60 and Hsp70, linking the disease pathogenesis to the cross-reactivity between Hsps from eukaryotic and prokaryotic organisms (Ciancio & Chang, 2008).

Recombinant L. lactis NCDO2118 producing Hsp65 protein was developed by our research group (de Azevedo et al., 2012). This recombinant strain showed promising results when it was tested in either colitis (25) and experimental autoimmune encephalomyelitis (EAE) (26) mice models. Thus, it was showed the immunomodulatory and anti-inflammatory action of Hsps molecules.  

Regarding the mechanisms associated with the beneficial effects of Hsp65, Gomes-Santos et al. (2017) are using the engineered L lactis mentioned above, associated the protection of the gut with increased Il10 levels in the colon and an expansion, in spleen and mesenteric lymph nodes, of CD4+Foxp3+ and CD4+LAP+ regulatory T cells, possibly in a dependent effect on Tlr2 and Il10. They also shown a reduction in pro-inflammatory cytokines and maybe, the same role could be observed in mucositis since both are injuries from the intestinal tract.

Methods: the authors should explain the reason why they used for assessing intestinal permeability the scintigraphy instead of test with sorbitol or mannitol.

The most common method for assessing permeability uses the principle of urinary excretion followed by oral administration of mannitol and lactulose sugars since they pass through the epithelium in a transcellular and paracellular form, respectively. However, this test is limited by the requirement of fasting, in addition to not being recommended for critically ill patients or those with kidney dysfunction (BJARNASON, MACPHERSON and HOLLANDER, 1995; KATOUZIAN et al., 2005; JORGENSEN et al., 2006).

An alternative is the use of radioactive elements such as 51Cr-EDTA and 99mTc-DTPA. These markers have the same physical properties as oligosaccharides and have the advantage of being easy to measure. 99mTc has a half-life of only 6 hours, allowing higher doses to be used, facilitating recovery in small blood or urine samples (SUN, WANG and ANDERSSON, 1998; JORGENSEN et al., 2006). Studies carried out by our research group have shown the effectiveness of using 99mTc-DTPA to determine intestinal permeability in an experimental model of intestinal obstruction (GENEROSO et al., 2010; SANTOS et al., 2010; VIANA et al., 2010, QUIRINO, 2012), prolonged physical exercise in a hot environment (COSTA, 2012), passive hyperthermia (SOARES, 2012) and ulcerative colitis (ANDRADE, 2013).

Moreover, were zonulin levels analyzed?

Levels of Zonulin itself were not evaluated directly (by ELISA, for example).

We did the Real-time (RT-qPCR) to detect and measurement of product levels. Quantitative reverse transcription PCR (RT-qPCR) is used when the starting material is RNA. In this method, RNA is first transcribed into complementary DNA (cDNA) by reverse transcriptase from total RNA or messenger RNA (mRNA). The cDNA is then used as the template for the qPCR reaction

For inflammation: were cytokine levels analyzed?

As we mentioned above, levels of cytokine itself were not evaluated directly by ELISA, for example. One of its major disadvantages is that the presence of RNA does not always accurately reflect protein levels. However, RT-qPCR is the standard golden method. That involves the measurement of cytokine mRNA transcript abundance. This method is relatively straightforward and quantitative and allows for detecting many different cytokines from relatively small sample amounts.

Study design: was the intestinal permeability and inflammation assessed even after L. delbrueckii (not hsp65) administration as a negative control? It is widely know that Lactobacilli themselves act reducing permeability inflammation, even in other settings (see and amend doi:10.3390/nu12092674)

We partially agree. Our group published (Inflamm Bowel Dis. 2012 Apr; 18 (4): 657-66. doi: 10.1002 / ibd.21834. Epub 2011 Aug 11) where we demonstrated that these activities are dependent strains. However, our strain has these activities proven in an article De Jesus et al (see below and in the manuscript on lines 105-109).

Thus, based on the beneficial action of Hsp65 protein in different diseases, and considering the protective effect of fermented milk by L. delbrueckii CIDCA 133 strain in an intestinal mucositis mouse model previously reported by De Jesus et al. [17]. The present study aim of to investigate the therapeutic/protective and immunomodulatory effect of recombinant L. delbrueckii CIDCA 133 (pExu:hsp65) on the deleterious effects of 5-FU in the intestinal epithelium.

Moreover, you are right, and our results confirm this fact. When you analyze the permeability and the histological score, the probiotic bacteria caring the empty plasmid has a beneficial effect; the group feeding with L. delbrueckii (pExu:hsp65) manifests a better effect as we expected.

RESULTS: The authors should conform figures and graphics style always using color figures and graphs.

Ok we understand, in the re-submission we will choose the color option. Thanks for the advice.

For points 3.7 and 3.8 the authors should report p values among values in parenthesis

We appreciate your colocation, and we add the p values among values in parenthesis. Please see the text below, and in the manuscript between the lines 483- 505.

3.7 rCIDCA 133: Hsp65 reduces gene expression of pro-inflammatory molecules and upregulates the IL10 expression in ileum of inflamed animals

The qPCR results reveal that the relative mRNA expression of Tnf (7.28±0.42), Il1b (2.3±0.49), Il6 (1.47±0.34), and Il12 (3.01±0.52) were up-regulated in animals from MUC group contrary to exhibited in the CTL group: Tnf (1.0±0.74) (p<0.0001), Il1b (1.0±0.14) (p<0.0001), Il6 (1.0±0.10) (p<0.05), Il12 (1.0±0.17) (p<0.0001). Oral treatment either rCIDCA 133 and rCIDCA 133: Hsp65, were able to suppress the Tnf (1.94±0.86; 1.48±0.13, respectively) (p<0.0001), Il1b (1.04±0.26; 0.40±0.11, respectively) (p<0.0001), Il6 (1.05±0.18; 0.22±0.07, respectively) (p<0.05; p<0.0001) and also Il12 (1.07±0.27; 1.56±0.75, respectively) (p<0.0001; p<0.001). It was observed that levels of the anti-inflammatory Il10 cytokine were reduced (0.08±0.03) (p <0.05) after the administration of 5-FU (MUC group) in relation to CTL group (1.0±0.27) (p<0.0001). Both recombinant strains studied rCIDCA 133 and rCIDCA 133: Hsp65 were able to up-regulate the Il10 (0.71±0.16; 0.77±0.20, respectively) (p<0.001), expression (Figure 6A-E).

3.8 Treatment with recombinant strains of Lactobacillus CIDCA 133 reduces MYD88, NOS2, TLR2 and TLR4 gene expression

MUC group showed a significant increase in mRNA expression of Myd88 (4.16±2.26), Nos2 (2.52±1.090), Tlr2 (2.32±1.08) and Tlr4 (3.26±1.58), when compared with the CTL group: Myd88 (1.00±0.14) (p<0.001), Nos2 (1.0±0.26) (p<0.001), Tlr2 (1.0±0.79) (p<0.01), and Tlr4 (1.0±0.40) (p<0.001)]. The treatment with both recombinant strains (rCIDCA133 and rCIDCA 133: Hsp65) resulted in decreased Myd88 (0.65±0.43; 0.03±0.02) (p<0.001; p<0.0001), Nos2 (0.34±0.14; 0.13±0.02) (p<0.0001), Tlr2 (0.77±0.20; 0.08±0.11) (p<0.05; p<0.0001) and Tlr4 (0.60±0.37; 0.03±0.01) (p<0.0001) gene expression, respectively. There were no statistical differences when these two recombinant groups were compared (Figure 6F-I). 

DISCUSSION: The authors should discuss the results of previous studies on the use of L. delbruecki (not hsp65 recombinant) on intestinal inflammation, permeability and more in general, in available, on human patients on 5-FU therapy.

We appreciate your suggestion. We add the following paragraph. Please see below and in the manuscript between lines 557 to 564.

The protective effect of fermented milk by wild type of Lactobacillus delbrueckii subsp. lactis CIDCA 133 strain was previously tested for the first time by our research group in a mucositis mice model (17). In this experiment, mice received fermented milk for 13 days, and at 10o day, they received the 5FU drug to induce the mucositis. Animal treated with this strain presented a reduction in the intestinal permeability values after the administering 99mTc-DTPA by gavage. These animals also exhibited preserved villus/crypt ratio, consequently showing a preserved epithelial architecture, with a significant amount of Goblet cells and reduction in inflammatory infiltrate. Altogether, shown the beneficial effect of this probiotic strain in intestinal inflammation, specifically in intestinal mucositis.

There are several typos throughout the manuscript that need correction

We did a complete revision.

Reviewer 2 Report

This study describes the protective effect of a recombinant LAB (expressing HSP65) in a model of 5-FU induced mucositis. The study is overall convincing, however some points should be clarified to support the authors claims. The most important issue is the fact the recombinant strain is not already published and there is no characterization in the present study of its ability to produce (and to secrete?) the HSP65 peptide. A general comment also, more care should be taken in writing the manuscript, there are a lot of typo mistakes, approximative abbreviations that do not fit with official nomenclature for genes/proteins.

I listed below major and minor points that need to be treated to allow consideration of this manuscript for publication:

Major points

1) Line 305-311: “3.1. Eukaryotic Cells are able to Express Hsp65 protein” Maybe I missed the point, but I do not see the interest of the first paragraph of results. The authors created a bacterial strain expressing Hsp65 but checked plasmid expression and localization of the corresponding proteins in transfected eukaryotic cells (cell line is not indicated by the way and there is no associated figure). Concerning the second part of this paragraph, if there is no data shown, this belongs to the material and methods section. Last, but the most important point, there is no clue in the paper that the recombinant bacterial strain produces and secretes the HSP65 peptide. Plasmid extraction and validation of the construction is not an evidence of its effective production by the bacteria. Western blot on whole bacterial lysate and maybe on the supernatant should be performed to validate the fact the recombinant strain produces the peptide of interest. Its production in eukaryotic cells is not a clue, this should be demonstrated in the bacteria. If HSP65 is not secreted by the recombinant bacteria it can challenge the conclusion of the paper. This should be clarif

2) Figure 4: Results demonstrating the loss of goblet cells and decrease in MUC2 expression are quite convincing, did they authors checked the consequences of this loss on the mucus layer. Is there differences in its thickness. Maybe it is not mandatory, but it could strengthen their conclusion.

3) If there is no objective quantification or scoring (as performed for histopathology data in figure 3 or 4, Figure 5G is not really convincing to support the statement of line 435: “The ultrastructural images of either recombinant treatments showed an improvement in all these analyzed parameters”. Is it possible to make a robust quantification supporting the statement ? alternatively immunostaining with antibodies raised against cell junctions proteins might allow a better quantification.

4) Figure 6: The authors said that they used the 2DDCt method to measure relative gene expression in their experimental groups. It means for that the average value for each tested genes should be equal to one for the control group (CTL), but in their graphs, values for the CTL group is highly variable from one target gene to another: TNF about 2, IL1B less than 1, IL-6 = 0.5, IL-10 = 1.6. In their settings, what is equal to 1 and serves as the reference value to express relative gene expression? I do not find the answer in the mat and meth section. This point should be clarified, and to my point of view results for the CTL group should be set at 1 for all target genes.

5) Figure 6: What is the reason to check expression of genes involved in the TLR pathway?  And NOS2? It lacks a brief explanation at the beginning of the paragraph to describe why this readout is relevant.

6) Table 2: results are difficult to appreciate, is possible to present them as histograms? Usually bacterial translocation is assessed by grinding the organs and plating the lysates on agar plates to count CFU number per gram of tissue, validating the viability of bacteria harvested from organs. How the authors could be sure that radioactivity measured is linked to viable bacteria translocated from the gut and not fragment of radiolabeled bacteria diffusing passively thanks to the increase permeability induced by the 5-FU (by the way are these bacteria are still alive with the concomitant erythromycin treatment?).  I am not convinced by this method to measure bacterial translocation.

7) Line 650-653 : “Our results are in accordance with others authors whose describe TLRs as the major sensors of the innate immune system which recognize highly preserved motifs of microorganisms 103, and related to the activation of signaling cascades, such as NFKB, associated to the inflammatory responses through the production of the pro-inflammatory cytokines 104; 105.” This sentence is overstated and should be removed, there is no clue that the higher expression of TLR-related genes observed in the MUC group is linked to the higher expression of pro-inflammatory-related genes.

Minor points:

Title and abstract: add the precision that it is a « mice » mucositis model (line 4 and 28), it is not clearly stated in the abstract.

Line 32: “and was observed an improvement in the histological scores on mice ileum”: not sure that the syntax of the sentence was right, rephrase if not.

Line 33: “TNF” : Is it TNF-α ? if yes, alpha should be added.

Line 34 and throughout the manuscript :” IL12 and IL1B, and increasing in IL-10” Correct names and abbreviations of proteins cited should be used: Line 34 and 281 : IL-12 is an heterodimer, is it IL-12A or B that was measured? IL12 is not the right abbreviation; line 34: Il1B is not the right short name for murin Interleukin-1 beta; line 281: typo mistake “Caludin” instead of Claudin. Please check the whole manuscript for proper names and abbreviations.

Line 36: “imagens” ? typo mistake?

Line 37: “the high gene expression in TLRs” syntax mistake

Line 54: “and customize the composition and activity of the gut microbiota” : Is it really the case ? now there are many studies describing few or no effects of probiotics on the gut microbiota composition, especially in human. Maybe this statement should be tempered?

Line 56:” The treatment of intestinal diseases, such as colitis and mucositis, with wild-type probiotics have 57 been reported showing promising results 14,15,16,17” : Except ref 14, the other references demonstrate effects of probiotics in mice models of colitis and mucositis. To be more rigorous, maybe the authors should precise that probiotics display promising results in model organisms models, but positive results are more limited in the treatment of human inflammatory diseases.

Line 60: “have been performed 18–20, 15,21, 22 » References are added in a confusing way… (true throughout the manuscript).

Line 62” Mycobacterium leprae” Italic

Line 63:” for prevention and treatment of several diseases such as tuberculosis 23,24, colitis 25, encephalomyelitis 26, lupus 27, atherosclerosis 28 among others, with excellent outcomes, t” If it is in animal models, please add this precision.

Line 313: to facilitate readers understanding it might be useful to precise in this first paragraph of result to which condition correspond the MUC group (e.g the one receiving 5-FU treatment).

Figure 1A: I understand that results appear more clearly with a y-axis starting at 35cm, but it is more rigorous to start it at 0 (similarly to all the other graphs of the paper).

Line 321: “The body weight loss of 321 mice of rCIDCA 133 and rCIDCA 133:Hsp65 groups were significantly lower (approximately 9% and 322 8%, respectively) than those in the MUC group (about 11%)” It is not what is indicated by your letters depicting statiscal analysis, letter b is present for the MUC and rCIDCA133, it means that difference is not significantly different between this two groups, right ? if so, text should be edited accordingly.

Line 343 “in MPO e EPO enzymes”: e = and ?

Line 369: Results paragraph 3.4 and 3.5 could be merged since they refer to the same figure 3 and the both present data regarding the integrity of the intestinal mucosa (or alternatively, and maybe it is better, figure 3A could be merged with figure 5 presenting tight junctions data).

Figure 3B: Inset are quite at the same magnification as the corresponding picture, if there is not a higher magnification allowing to see greater details they are useless and can be removed.

Figure 3F: Ratios have no units, thus (µm) should be removed from the y-axis.  

Figure 5A: “Occludina” : “a” at the end should be removed

Paragraph 3.7 and 3.8: please check if the proper abbreviation is used for genes/proteins. For example, gene name of TNF-alpha is Tnf or for the protein TNF-a, but not TNFA. As asked above, is IL12, is IL12A or IL12B?

Line 489 and 502 :”M leprae” : italic

Line 504: “we decide to develop a DNA vaccine using the pExu vector” is it really a vaccine that is developed here ? or a plasmid ?

Line 505: “After confirming the functionality of pExu:hsp65 vector in eukaryotic cells,” Useless to my point of view, if the authors want to create a bacterial strain expressing Hsp65, there is no interest to validate the expression in eukaryotic host cells.

Line 536: “architect” : architecture

Line 609: “All the tight junction proteins studied, except Claudin 5, increase significantly the mRNA 610 expression in recombinant treatment.” Confusing sentence, please rephrase.

Line 647-649 :” this information could explain our results regarding the high translocation observed in animals treated with rCIDCA: Hsp65, since they presented extremely lower levels, even more than the negative control, of TLR4 mRNA expression and also MYD88.” TLRs signaling is complex, thus additional experiments to measure TLRs protein levels, TLRs protein localization (especially in polarized cells such as enterocytes), or assessing postranslationnal modifications such as phosphorylations that govern signaling downstream TLRs are necessary to investigate possible effects on these signaling pathway. mRNA analysis is not a sufficient readout to conclude on effects on TLR signaling. A sentence to notify this could be added in the discussion section.

Line 659: IBD  : use the full length name since there is no other occurrence of inflammaroty bowel diseases in the manuscript.

Line 673-674 “Thus, the recombinant probiotic L. delbrueckii CIDCA 133 (pExu:hsp65) consumption, shows to be a good alternative to ameliorating the intestinal damage causing by 5-FU” Add “in mice model”.

Author Response

Dear Dr,

Following, our answers regarding your appontiments.

This study describes the protective effect of a recombinant LAB (expressing HSP65) in a model of 5-FU induced mucositis. The study is overall convincing, however some points should be clarified to support the authors claims. The most important issue is the fact the recombinant strain is not already published and there is no characterization in the present study of its ability to produce (and to secrete?) the HSP65 peptide. A general comment also, more care should be taken in writing the manuscript, there are a lot of typo mistakes, approximative abbreviations that do not fit with official nomenclature for genes/proteins.

We appreciate your appointments, but it is necessary to clarify that, in this specific case [L. delbrueckii CIDCA 133 (pExu:hsp65)], L delbrueckii does not act as a machine for heterologous protein production. Here, we used another approach, the pExu vector, which was developed for DNA vaccine (Mol Ther Methods Clin Dev. 2016 Dec 24;4:83-91. doi: 10.1016/j.omtm.2016.12.005. eCollection 2017 Mar 17). A New Broad Range Plasmid for DNA Delivery in Eukaryotic Cells Using Lactic Acid Bacteria: In Vitro and In Vivo Assay and Front Microbiol.. 2018 Oct 5;9:2398. doi: 10.3389/fmicb.2018.02398. eCollection 2018. Microencapsulation of Lactic Acid Bacteria Improves the Gastrointestinal Delivery and in situ Expression of Recombinant Fluorescent Protein)

We perform the transfection of eukaryotic cells (exactly to demonstrate that these cells were able to produce and express the hsp65 protein) and in vivo experiment. We add a figure (Figure 1, lines 345-351) which demonstrates the results of the transfection assay.

I listed below major and minor points that need to be treated to allow consideration of this manuscript for publication:

Major points

1) Line 305-311: “3.1. Eukaryotic Cells are able to Express Hsp65 protein” Maybe I missed the point, but I do not see the interest of the first paragraph of results. The authors created a bacterial strain expressing Hsp65 but checked plasmid expression and localization of the corresponding proteins in transfected eukaryotic cells (cell line is not indicated by the way and there is no associated figure). Concerning the second part of this paragraph, if there is no data shown, this belongs to the material and methods section. Last, but the most important point, there is no clue in the paper that the recombinant bacterial strain produces and secretes the HSP65 peptide. Plasmid extraction and validation of the construction is not an evidence of its effective production by the bacteria. Western blot on whole bacterial lysate and maybe on the supernatant should be performed to validate the fact the recombinant strain produces the peptide of interest. Its production in eukaryotic cells is not a clue, this should be demonstrated in the bacteria. If HSP65 is not secreted by the recombinant bacteria it can challenge the conclusion of the paper. This should be clarif

We appreciate your appointment, but this approach was used to test a DNA vaccine platform, as explained in the previous answer. In this case, the most important thing is the host cells being able to express the protein.

Otherwise, the plasmid vector is unfunctional. Western blot is not necessary to be mandatory for eukaryotic cell transfection and immunohistochemical assays to demonstrate the HSP protein expression by eukaryotic cells. We performed this experiment and we confirmed the expression. If you consider it necessary, we can add this information on the manuscript. To clarify, we add some of these points in the text. Please see lines 102-104. Please see the following text.

Based on these promising results for heterologous protein production by L. lactis, our research group developed a broad range plasmid called pExu (doi: 10.3389/fmicb.2018.02398) to be used as a DNA vaccine vector, in this approach, protein production is in charge of the host cells, being LAB the delivery vehicle.

2) Figure 4: Results demonstrating the loss of goblet cells and decrease in MUC2 expression are quite convincing, did they authors checked the consequences of this loss on the mucus layer. Is there differences in its thickness? Maybe it is not mandatory, but it could strengthen their conclusion.

Unfortunately, we cannot perform this analyze. In the histological processing, the point is that due to the reagents used, there is a loss in the mucus layer losing its continuity. Thus, we do not consider it appropriated to make this measurement because it could lead to unreal results. However, we appreciate your appointment.

3) If there is no objective quantification or scoring (as performed for histopathology data in figure 3 or 4, Figure 5G is not really convincing to support the statement of line 435: “The ultrastructural images of either recombinant treatments showed an improvement in all these analyzed parameters”. Is it possible to make a robust quantification supporting the statement? alternatively immunostaining with antibodies raised against cell junctions’ proteins might allow a better quantification.

Performing the electronic microscopy, we only want to show that we have a better or worst organization (depending on the treatment) of the thigh junctions. As we could see, the junction proteins’ gene expressions corroborate with the EM figure’s findings. We change the sentence, as you suggested. Please see lines 477-479, or the following sentence.

The ultrastructural analyses for either recombinant treatments corroborate with the findings of relative gene expression showing, at least in part, that these parameters were ameliorated in animals which received both recombinant strains.

We know that the experiment you suggest could came to enrich our results. However, the obtained results are enough to show the recombinant strain, carrying the hsp65 gene, has a beneficial effect in mucositis. Furthermore, is important to highlight that we are not able to perform these analyzes because our University is still closed and only researches regarding SARS-COV2 are allowed, which is not the case.

4) Figure 6: The authors said that they used the 2DDCt method to measure relative gene expression in their experimental groups. It means for that the average value for each tested genes should be equal to one for the control group (CTL), but in their graphs, values for the CTL group is highly variable from one target gene to another: TNF about 2, IL1B less than 1, IL-6 = 0.5, IL-10 = 1.6. In their settings, what is equal to 1 and serves as the reference value to express relative gene expression? I do not find the answer in the mat and meth section. This point should be clarified, and to my point of view results for the CTL group should be set at 1 for all target genes. 

Yes, you are right regarding 2DDCt. We fixed it, the result to CTL were set at 1. Thanks for the guidance.

5) Figure 6: What is the reason to check expression of genes involved in the TLR pathway?  And NOS2? It lacks a brief explanation at the beginning of the paragraph to describe why this readout is relevant.

We consider your appointment very relevant; we are so glad for that. We add the following information on the manuscript. Please read the paragraph below, and it is inserted in the manuscript between lines 697-710.

As 5-FU generates a dysbiosis in the bowel, disruption of the mucosal barrier upon injury to intestinal epithelial cells creates Tlr ligands’ exhibition by commensal bacteria to Tlr expressing in many bowel cells (Strober et al 2002). Thus, due to the significant role of the of microbiota in intestinal inflammation, the active participation of Tlr in the bacterial product recognition and their importance in the inflammation induction encourages us to test the in vivo effect of Tlr ligation by commensal-derived products. Other essential intracellular adapters as Myd88 and Nod2 were also investigated. Tlrs (except Tlr3) stimulate the cells through the MyD88, which mediates the early immune response to pathogens, leading to Nfkb translocation to the nucleus and consequently, genes expression of encoding pro­inflammatory cytokines and chemotactic cytokines (chemokines) (Takeda et al., 2003; O’Neill et al., 2013). The microbial product also can be recognized by a family of intracellular signaling proteins, called Nod. They can detect microbial ligands in the cytosol. These cytosolic proteins signal activation of Nfkb and MAPK, followed by induction of numerous genes involved in the inflammatory process, thus trigger host innate immune responses (Inohara et al 2000; Ogura et al 2001; Hayden and Ghosh et al., 2004). Nod2 recognizes muramyl dipeptide on degraded bacterial cell wall peptidoglycan and can therefore respond to invading Gram ­negative and Gram ­positive bacteria.

6) Table 2: results are difficult to appreciate, is possible to present them as histograms?  

As your suggestion, we change the table for a histogram. There was a noticeable improvement. Now, data are presented as figure 8.

Usually bacterial translocation is assessed by grinding the organs and plating the lysates on agar plates to count CFU number per gram of tissue, validating the viability of bacteria harvested from organs. How the authors could be sure that radioactivity measured is linked to viable bacteria translocated from the gut and not fragment of radiolabeled bacteria diffusing passively thanks to the increase permeability induced by the 5-FU (by the way are these bacteria are still alive with the concomitant erythromycin treatment?).  I am not convinced by this method to measure bacterial translocation.

As we know and many papers have been shown the failure of the intestinal barrier function, resultant in an increase of mucosal permeability, and consequently is a significant promoter of Bacterial Translocation.

As you mention, the traditional method is the assayed based on an agar plate to count the CFU number per gram of tissue. However, this method has many drawbacks, such as time. Some bacteria grow more prolonged than other; specificity, to perform an accurate assay, is better to plate the macerated organs in more than one medium not to lose any spice. All the translocated bacteria do not gown in the medium that you are testing, or maybe they grew at different times and in different oxygen and temperature conditions.  Consequently, the counted bacteria could be lower than the real translocated once. Thus, the obtained results could be underestimate, and consequently, the assay loss accuracy.

This method only tests viable bacteria, and in our opinion, it is a significant drawback because many PAMPS are possible to translocate and could contribute to the persistently elevated systemic levels of PAMPs, which is not desirable. This point is crucial because the definition of BT is not only the passage of viable resident bacteria, but is the passage of viable resident bacteria and inert particles, and other macromolecules such as PAMPs, across the intestinal mucosal barrier to mesenteric lymph nodes and other internal organs (Woodcock et al 2001; Gatt et al 2002; Mac Fie et al 2004). Furthermore, complemented with exciting information from doi:10.1016/j.jim.2014.12.015, where the authors analyze different methods to evaluate permeability and bacterial translocation, they suggested  “that an ideal permeability assay will use labeled probes (such as bacteria, proteins or macromolecules). However, no universal marker provides a definitive answer on the leakiness of the intestine, and a combination of methods assessing the flow from the blood to the intestinal lumen might be useful.”

We are showing that the 99mTc-DTPA is a very good method to test both permeability and also bacteria translocation.

Regarding the erythromycin, we use it because we are working with recombinant bacteria, which need the selection marker to grow. To study the control group is vital to keep the same background (as we use erythromycin to select the recombinant bacteria, we also add this antibiotic in the medium for the other experimental groups, so all groups were exposed to the same conditions). We use very low concentration (2.5ug/mL), and also, after two days from the beginning of the experiment, also in the middle and at the end of that, we platted fezzes samples in three different mediums, and we saw a considerable number of bacteria in all of tested medium.

Gatt M, Reddy BS, MacFie J. Review article: bacterial translocation in the critically ill-evidence and methods of prevention. Aliment Pharma- col Ther 2007;25:741–57.

Mac Fie J.Current status of bacterial translocation as a cause of surgical sepsis. Br Med Bull 2004;71:1–11.

Woodcock NP, Robertson J, Morgan DR, Gregg KL, Mitchell CJ, MacFie J. Bacterial translocation and immunohistochemical measurement of gut immune function. J Clin Pathol 2001;54:619–23.

7) Line 650-653 : “Our results are in accordance with other authors whose describe TLRs as the major sensors of the innate immune system which recognize highly preserved motifs of microorganisms 103, and related to the activation of signaling cascades, such as NFKB, associated to the inflammatory responses through the production of the pro-inflammatory cytokines 104; 105.” This sentence is overstated and should be removed, there is no clue that the higher expression of TLR-related genes observed in the MUC group is linked to the higher expression of pro-inflammatory-related genes.

We agree with you and apologize for that. We add a sentence at the end of this paragraph as follow. Please see below and between lines 716-718 on the manuscript.

This analysis shows, at least, a potential relationship between these parameters. However, mRNA expression cannot affirm that different circulant cytokines' different concentrations will be found overdue to the possible post-translational modification.

Minor points:

Title and abstract: add the precision that it is a « mice » mucositis model (line 4 and 28), it is not clearly stated in the abstract.

Ok. The changes were done

Line 32: “and was observed an improvement in the histological scores on mice ileum”: not sure that the syntax of the sentence was right, rephrase if not.

Yes, really the phrase was a little confusing we change it as you can see in the following paragraph. Please see lines 33 to 35.

“This approach showed increased levels of sIgA in the intestinal fluid, reduction in both inflammatory infiltrate and intestinal permeability. Also, the histological score was improved.”

Line 33: “TNF” : Is it TNF-α ? if yes, alpha should be added.

Yes, however, we adopted the new nomenclature found in http://www.informatics.jax.org/

Line 34 and throughout the manuscript :” IL12 and IL1B, and increasing in IL-10” Correct names and abbreviations of proteins cited should be used: Line 34 and 281 : IL-12 is an heterodimer, is it IL-12A or B that was measured? IL12 is not the right abbreviation; line 34: Il1B is not the right short name for murin Interleukin-1 beta; line 281: typo mistake “Caludin” instead of Claudin. Please check the whole manuscript for proper names and abbreviations.

We checked all the abbreviations, and now, throughout the manuscript, the abbreviations follow the recommendations found in http://www.informatics.jax.org/.

Line 36: “imagens” ? typo mistake?

Yes, it was a typo mistake. We concerted it.

Line 37: “the high gene expression in TLRs” syntax mistake.

 We apologize for that. We fix the sentence ……reverse the high levels in gene expression of TLRs caused….

Line 54: “and customize the composition and activity of the gut microbiota” : Is it really the case ? now there are many studies describing few or no effects of probiotics on the gut microbiota composition, especially in human. Maybe this statement should be tempered?

Yes, you are right, we add the following paragraph. Please see in the manuscript between lines 64 to 70.

In this context, researches have been developed using the microbial 65 kDa Heat Shock Protein (Hsp65) of Mycobacterium leprae (homolog to mammalian Hsp60) in many different animal disease models, to evaluate either prevention or treatment. For instance, to evaluate the effect of this protein in tuberculosis disease, mice [23] and calves [24] were used. Mice model were used to study colitis [25], encephalomyelitis [26], lupus [27], atherosclerosis [28] among others, with excellent outcomes, thus demonstrating the relevance of the Hsp65 protein as a good candidate for treatment and therapeutic uses.

Line 56:” The treatment of intestinal diseases, such as colitis and mucositis, with wild-type probiotics have 57 been reported showing promising results 14,15,16,17” : Except ref 14, the other references demonstrate effects of probiotics in mice models of colitis and mucositis. To be more rigorous, maybe the authors should precise that probiotics display promising results in model organisms models, but positive results are more limited in the treatment of human inflammatory diseases.

Yes, you are right, we add the following paragraph. Please see below, and the manuscript lines 64-70

In this context, researches have been developed using the microbial 65 kDa Heat Shock Protein (Hsp65) of Mycobacterium leprae (homolog to mammalian Hsp60) in different animal disease models, to evaluate either prevention or treatment. For instance, to evaluate the effect of this protein in tuberculosis disease, mice [23] and calves [24] were used. Mice model were used to study colitis [25], encephalomyelitis [26], lupus [27], atherosclerosis [28] among others, with excellent outcomes, thus demonstrating the relevance of the Hsp65 protein as a good candidate for treatment and therapeutic uses.

Line 60: “have been performed 18–20, 15,21, 22 » References are added in a confusing way… (true throughout the manuscript).

Yes, we check and link again all the references again.  We apologize for this mistake.

Line 62” Mycobacterium leprae” Italic.

OK we fix it.

Line 63:” for prevention and treatment of several diseases such as, tuberculosis (23), (24); colitis 25, encephalomyelitis 26, lupus 27, atherosclerosis 28 among others, with excellent outcomes, t” If it is in animal models, please add this precision.

The required changes were done. We changed the paragraph a little. Please at the manuscript, see lines 66-70.

For instance, to evaluate the effect of this protein in tuberculosis disease, mice [23] and calves [24] were used. Mice model were used to study colitis [25], encephalomyelitis [26], lupus [27], atherosclerosis [28] among others, with excellent outcomes, thus demonstrating the relevance of the Hsp65 protein as a good candidate for treatment and therapeutic uses.

Line 313: to facilitate readers understanding it might be useful to precise in this first paragraph of result to which condition correspond the MUC group (e.g the one receiving 5-FU treatment).

All the details are in material and methods. If we add this information in the results section, this information will be disconnected, besides being repetitive. 

Figure 1A: I understand that results appear more clearly with a y-axis starting at 35cm, but it is more rigorous to start it at 0 (similarly to all the other graphs of the paper).

This alteration was done. Please see a new 2A figure.

Line 321: “The body weight loss of 321 mice of rCIDCA 133 and rCIDCA 133:Hsp65 groups were significantly lower (approximately 9% and 322 8%, respectively) than those in the MUC group (about 11%)” It is not what is indicated by your letters depicting statiscal analysis, letter b is present for the MUC and rCIDCA133, it means that difference is not significantly different between this two groups, right ? if so, text should be edited accordingly.

Yes, there were some mistakes. We edited the text. We apologize for these mistakes. Please see the following paragraph, and lines 360-362 on the manuscript.

The time-course of the mice weight was another parameter evaluated. The body weight loss of mice of rCIDCA 133:Hsp65 group was significantly lower (approximately 4 %) than those in the MUC group (about 5.5%) (p<0.01) (Figure 1B). No mortality was observed during the experiment.

Line 343 “in MPO e EPO enzymes”: e = and?  

Yes, you are right. We apologize for not seeing this detail.

Line 369: Results paragraph 3.4 and 3.5 could be merged since they refer to the same figure 3 and the both present data regarding the integrity of the intestinal mucosa (or alternatively, and maybe it is better, figure 3A could be merged with figure 5 presenting tight junctions data).

We have analyzed your suggestion, which we consider to be excellent. However, when we reread it, we saw a lot of information that would have to be concentrated and that it could be difficult for the reader to understand. We made some merger attempts, and we didn’t like the result. We want to know if it is possible to leave it because it facilitates the reading and interpretation of researchers from our point of view.

Figure 3B: Inset are quite at the same magnification as the corresponding picture, if there is not a higher magnification allowing to see greater details they are useless and can be removed.

Yes, we agree with your advice and withdraw the zoom. Please see figure 4B.

Figure 3F: Ratios have no units, thus (µm) should be removed from the y-axis.  

Ok, we change the legend on y-axis. We correct the mistake, and we apologize for that.

Figure 5A: “Occludina” : “a” at the end should be removed

OK

Paragraph 3.7 and 3.8: please check if the proper abbreviation is used for genes/proteins. For example, gene name of TNF-alpha is Tnf or for the protein TNF-a, but not TNFA. As asked above, is IL12, is IL12A or IL12B?

As we mentioned above, now all the abbreviations are following

http://www.informatics.jax.org/

Line 489 and 502 :”M leprae” :

Italic Ok, we put the bacteria name in italic form; the mistake was made, when we put the manuscript in the journal template. We apologize for not seeing this detail.

Line 504: “we decide to develop a DNA vaccine using the pExu vector” is it really a vaccine that is developed here ? or a plasmid ?

In fact, in this work, we develop the DNA vaccine L. delbrueckii CIDCA 133 (pExu:hsp65). The plasmid was developed in earlier work. Here is the doi of this research (doi:10.1016/j.omtm.2016.12.005.).

Line 505: “After confirming the functionality of pExu:hsp65 vector in eukaryotic cells,” Useless to my point of view, if the authors want to create a bacterial strain expressing Hsp65, there is no interest to validate the expression in eukaryotic host cells.

Previously we developed the pExu vector (DOI 10.1016/j.omtm.2016.12.005). pExu is a vector with a mammalian promoter, the CMV. For that reason, we do not expect protein expression by the bacteria. Our proposal is the BL transfer/deliver the plasmid into eukaryotic cells. In this condition, these cells are in charge of protein expression. We add a little sentence to explain this in the introduction to avoid confusion. We apologize for that. Please see the following sentences, and please see below and manuscript lines 102 to 104.

Based on these promising results for heterologous protein production by L. lactis, our research group developed a broad range plasmid called pExu (48) to be used as a DNA vaccine vector. In this approach, protein production is in charge of the host cells, being LAB the delivery vehicle.

Line 536: “architect”: architecture,

Ok the change was done

Line 609: “All the tight junction proteins studied, except Claudin 5, increase significantly the mRNA 610 expression in recombinant treatment.” Confusing sentence, please rephrase.

Please see the text in red. The sentence was changed. We improved it. Please, see the following sentences. They are in lines 654 to 656.

When the mRNA expression of tight junctions in the ileum section, were analyzed, an increase was observed. However, gene expression of Claudin 5 did not show increased mRNA expression.

Line 647-649 :” this information could explain our results regarding the high translocation observed in animals treated with rCIDCA: Hsp65, since they presented extremely lower levels, even more than the negative control, of TLR4 mRNA expression and also MYD88.” TLRs signaling is complex, thus additional experiments to measure TLRs protein levels, TLRs protein localization (especially in polarized cells such as enterocytes), or assessing postranslationnal modifications such as phosphorylations that govern signaling downstream TLRs are necessary to investigate possible effects on these signaling pathway. mRNA analysis is not a sufficient readout to conclude on effects on TLR signaling. A sentence to notify this could be added in the discussion section.

As your suggestion, a sentence was added to clarify the idea. Please see below and on the manuscript lines 694-696.

However, further experiments should be performed to dilucidated the mechanisms of this complex signaling pathway.

Line 659: IBD: use the full length name since there is no other occurrence of inflammaroty bowel diseases in the manuscript.

Ok. The correction was done.

Line 673-674 “Thus, the recombinant probiotic L. delbrueckii CIDCA 133 (pExu:hsp65) consumption, shows to be a good alternative to ameliorating the intestinal damage causing by 5-FU” Add “in mice model”.

As the reviewer suggested, the phrase in the mice model was added. Please see in the manuscript in line 733.

Round 2

Reviewer 1 Report

The authors improved their manuscript with the comments to my question, which is now ready for publication.

Reviewer 2 Report

Globally, the authors satisfied my concerns, thank you for the clarification regarding the pExu vector.